# Human mitochondria require mtRF1 for translation termination at non-canonical stop codons

Annika Krüger [1,2], Cristina Remes[3], Dmitrii Igorevich Shiriaev[1,2], Yong Liu [1,2], Henrik Spåhr[1,2], Rolf Wibom[4], Ilian Atanassov [5], Minh Duc Nguyen[1,2], Barry S. Cooperman[3] & Joanna Rorbach [1,2,6] ✉

The mitochondrial translation machinery highly diverged from its bacterial counterpart. This includes deviation from the universal genetic code, with AGA and AGG codons lacking cognate tRNAs in human mitochondria. The locations of these codons at the end of *COX1* and *ND6* open reading frames, respectively, suggest they might function as stop codons. However, while the canonical stop codons UAA and UAG are known to be recognized by mtRF1a, the release mechanism at AGA and AGG codons remains a debated issue. Here, we show that upon the loss of another member of the mitochondrial release factor family, mtRF1, mitoribosomes accumulate specifically at AGA and AGG codons. Stalling of mitoribosomes alters COX1 transcript and protein levels, but not ND6 synthesis. In addition, using an in vitro reconstituted mitochondrial translation system, we demonstrate the specific peptide release activity of mtRF1 at the AGA and AGG codons. Together, our results reveal the role of mtRF1 in translation termination at non-canonical stop codons in mitochondria.

Although the basic mechanisms of protein synthesis are conserved across all kingdoms of life, translational machineries have developed distinctive features throughout evolution. Eukaryotic cells contain two separate protein synthesis machineries: one in the cytosol and one in mitochondria, which are required to be highly coordinated for cellular homeostasis. Mitochondria are essential semi-autonomous eukaryotic organelles deriving from an alpha-proteobacterial lineage, and their translation mechanisms are similar to those of their bacterial counterparts. However, throughout evolution, the mitochondrial translational machinery has acquired specific features, including structural and functional specialization of the mitochondrial ribosomes (mitoribosomes), translation factors, tRNAs and mRNAs, and a deviation from the so-called universally conserved codon usage. In human

mitochondria, the standard stop codon UGA is reassigned to tryptophan and the codons AUA/AUU encoding isoleucine are reassigned to start codons. In addition, the codons AGA and AGG, which encode arginine according to the universal code, have no cognate tRNAs in mitochondria. These codons are located at the ends of the mitochondrial open reading frames (mtORFs) *COX1* and *ND6*, respectively (Fig. 1a), and were therefore initially assigned as stop codons[1]. Later, an alternative translation termination mechanism at these codons was suggested, according to which a −1 frameshift places a standard stop codon (UAG) in the mitoribosomal A-site[2]. While this hypothesis might explain the release of COX1 and ND6 in human mitochondria, it is not compatible with several vertebrate species having AGA/AGG codons that are not preceded by U, such that a −1 frameshift would not place a

[1]Department of Medical Biochemistry and Biophysics, Division of Molecular Metabolism, Karolinska Institutet, Biomedicum, 171 65 Solna, Sweden. [2]Max Planck Institute Biology of Ageing - Karolinska Institutet Laboratory, Karolinska Institutet, Stockholm, Sweden. [3]Department of Chemistry, University of Pennsylvania, Philadelphia, PA 19104, USA. [4]Department of Laboratory Medicine, Karolinska Institutet, Stockholm, Sweden. [5]Proteomics Core Facility, Max-Planck-Institute for Biology of Ageing, Joseph-Stelzmann-Str. 9b, 50931 Cologne, Germany. [6]S.T.I.A.S: Stellenbosch Institute for Advanced Study, Marais Rd, Mostertsdrift, Stellenbosch 7600, South Africa. ✉e-mail: Joanna.rorbach@ki.se

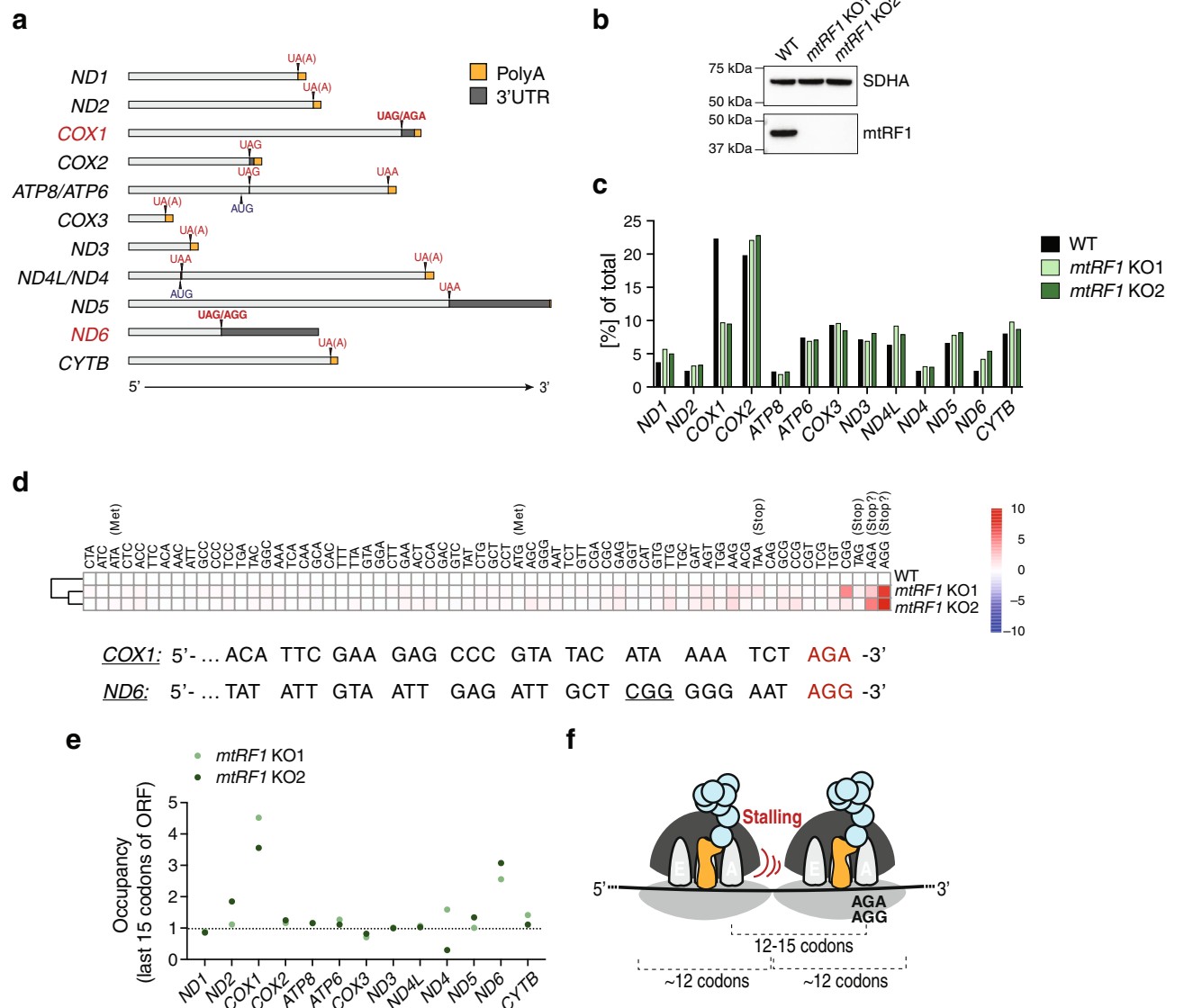

**Fig. 1 | Mitoribosomes stall at the ends of *COX1* and *ND6* transcripts in *mtRF1* KO cells. a** Schematic representation of human mitochondrial transcripts including polyA tails (orange), 3′ untranslated regions (3′UTRs) (dark gray), stop codons (red), and start codons of bicistronic transcripts (dark blue). Relative lengths of regions are depicted. **b** Western blot confirming the generation of *mtRF1* KO cells. 40 μg mitolysates were loaded. Two different clones are shown (KO1 and KO2). Loading was assessed by SDHA detection. A representative blot of *n* = 3 independent experiments is depicted. **c–e** Analysis of ribosome profiling data of WT and *mtRF1* KO cells. **c** Occupancy of mitoribosomes on each transcript relative to the total occupancy on all transcripts. **d** Upper panel: Codon counts relative to WT cells. Codons are organized according to sequencing coverage (high (left) to low number of reads (right)). Lower panel: Sequences of *COX1* and *ND6* transcript ends. Of note, codon CGG (underlined) occurs only once in human mitochondrial transcripts. **e** Occupancy of mitoribosomes on the last 15 codons of each transcript relative to the total occupancy on the respective transcript. Fold change values relative to WT cells are shown. Occupancies on each individual codon are presented in Supplementary Fig. 4. **f** Schematic representation of stalling events 12–13 codons upstream of stalling on AGA/AGG codons. Ribosome-protected fragments (RPFs) of mitoribosomes usually comprise 35 nucleotides (-12 codons).

standard stop codon in the ribosomal A-site[3]. Thus, whether the codons AGA and AGG represent functional stop codons in human mitochondria remains a highly debated issue[2–14].

Four proteins with known or putative peptide release activity localizing to mitochondria have been identified so far: mtRF1a, mtRF1, C12orf65, and ICT1. MtRF1a was shown to terminate the translation of at least 11 of the 13 mitochondrially encoded ORFs at the UAA/UAG standard stop codons[7,8,10]. C12orf65 and ICT1 lack a codon-recognition domain and contain positively charged C-terminal extensions, similar to the bacterial ribosome rescue factors[9,10,15]. ICT1 has been shown to have peptide release activity on mitoribosomes programmed with short mRNAs and containing the AGA/AGG stop codons in the A-site[9]. However, since its activity is greatly reduced when the mRNA exceeds

14 nucleotides past the ribosomal A-site[16] and a recent cryo-EM study revealed the binding of ICT1 to the empty mRNA channel[10], it is unlikely that ICT1 functions as a canonical release factor in vivo. MtRF1 was first identified in 1998 by a database search and suggested to be a classical release factor[17]. It likely arose by gene duplication from *MTRF1A*, with which it has very high sequence homology. Both proteins have a codon-recognition domain, which allows specific recognition of stop codons, and a GGQ domain, which catalyzes peptidyl-tRNA hydrolysis. Yet, mtRF1 displays distinct structural features, and its function has remained elusive. First, it contains alterations in the helix α5 region and the recognition loop, which were suggested to allow recognition of the stop codons AGA/AGG instead of UAG/UAA[12], or to bind to stalled mitoribosomes with empty A-sites[18]. Second, it harbors an extended

N-terminal domain, whose function is not understood. In bacterial RF2, the N-terminal domain contacts the ribosome[19], potentially stabilizing binding to the ribosome and/or supporting its right positioning. Interestingly, the appearance of mtRF1 coincides with the appearance of AGA/AGG at the root of the vertebrate lineage[3]. Nevertheless, so far all biochemical assays failed to show the release activity of mtRF1 on any of the mitochondrial termination codons[7-9]. Furthermore, a recent cryo-EM screening of mitochondrial release factors identified mtRF1a bound to ribosomal complexes programmed with the standard stop codons UAG and UAA in the A-site, but it did not detect any of the mitochondrial release factors bound to ribosomal complexes programmed with AGA and AGG[10].

Here, we show that upon loss of mtRF1, mitoribosomes stall at AGA and AGG codons, which affects COX1 transcript and protein levels, but not ND6 synthesis. In addition, our in vitro mitochondrial translation assays, shows the release activity of mtRF1 at both AGA and AGG, confirming that mtRF1 is involved in translation termination.

## Results

### Loss of mtRF1 leads to stalling of mitoribosomes at AGG and AGA stop codons

Despite numerous studies on translation termination in mammalian mitochondria, the function of the mitochondrial release factor mtRF1 has remained elusive. To study the molecular role of mtRF1, we generated *mtRF1* KO HEK293 cell lines by plasmid-based CRISPR/Cas9 delivery. Two clones were selected for consecutive experiments (*mtRF1* KO1 and KO2), and loss of mtRF1 was confirmed by western blotting (Fig. 1b) and sequencing (Supplementary Fig. 1a). Neither of the two clones showed a growth defect in galactose medium (Supplementary Fig. 1b), ruling out a general mitochondrial defect upon loss of mtRF1. Furthermore, neither mitoribosomal protein (MRP) levels (Supplementary Fig. 1c and Supplementary Table 3) nor the integrities of mitochondrial small subunits (mtSSUs), large subunits (mtLSUs) and monosomes (Supplementary Fig. 1d) were substantially affected by loss of mtRF1, suggesting no direct effect of mtRF1 on mitoribosome stability. Next, we analyzed mitoribosome stalling events upon loss of mtRF1 by ribosome profiling[20] and examined the relative occupancy of mitoribosomes on the 13 mtORFs (Fig. 1c and Supplementary Fig. 2a). Interestingly, while we observed similar mitoribosome occupancies in the HEK293 wildtype (WT) and *mtRF1* KO cells for most of the transcripts, occupancy on COX1 was decreased from 20% of total mitoribosome protected fragments (mtRPFs) in WT cells to only 10% of mtRFPs in *mtRF1* KO cells. As measured by relative occupancies, a decrease in ribosome occupancy on one of the transcripts is expected to cause a general increase in occupancies on other transcripts. Accordingly, we observed a slight increase in occupancies for most transcripts in *mtRF1* KO cells compared to the WT cells. Yet, we observed a significantly larger increase of mtRPFs (approximately two times) on the ND6 transcript.

These results indicate that loss of mtRF1 has an influence on either the transcript levels of COX1 and ND6, or on their translation rates.

Next, we analyzed ribosome occupancies with single codon resolution to identify potential stalling events at specific codons (Fig. 1d). Strikingly, we observed an enrichment of mitoribosomes at AGA and AGG codons in both *mtRF1* KO cell lines compared to WT cells. Additionally, we observed a significant enrichment at the codon CGG in KO1 cells. This codon occurs only once in all mitochondrial transcripts, at the end of ND6 (Fig. 1d, lower panel). Thus, our results indicate that ribosomes are indeed stalled at the ends of COX1 and ND6 transcripts in *mtRF1* KO cells, supporting the hypothesis that mtRF1 releases at AGA and AGG codons.

To obtain deeper insights into the mitoribosome stalling events upon loss of mtRF1, we analyzed mitoribosome occupancies at the last 15 codons upstream of the stop codons in each transcript (Fig. 1a, e). To compensate for different ribosomal distributions on transcripts in *mtRF1* KO cells compared to WT cells, mitoribosome occupancies at the end of each ORF were normalized to the total occupancy on the corresponding ORF. Confirming our previous findings, mtRPFs in *mtRF1* KO cells were enriched at the end of COX1 (4.5- and 3.5-fold) and ND6 (2.5- and 3.0-fold) mtORFs. Pronounced stalling events were also observed 12–15 codons upstream of AGA and AGG codons (Fig. 1f and Supplementary Fig. 2b). This result is in accordance with previous ribosome profiling experiments in which depletion of eRF1 resulted in queuing of ribosomes upstream of the stop codons[21]. The upstream stalling events were more numerous than stalling at AGA/AGG codons, as observed by Wu et al.[21]. This effect is likely due to variations in footprint lengths. The absence of release factors causes accumulation of stalled ribosomes with an empty A-site, which are more efficiently degraded during RNase treatment and result in shorter footprints[21]. As shorter footprints were not included in our analysis, we might have underestimated the stalling events at AGA/AGG codons. Other than stalling on COX1 and ND6 transcript ends, no significant differences were observed between WT and *mtRF1* KO cells.

Taken together, our data suggest that mtRF1 functions as a release factor at the COX1 (AGA) and ND6 (AGG) mtORFs, since loss of mtRF1 results in pronounced stalling events at the ends of both transcripts.

### Stalling at AGA/AGG has different effects on COX1 and ND6 transcript levels and translation

Our mitoribosome profiling data showed that mtRPFs decrease on COX1 transcripts and increase on ND6 transcripts in *mtRF1* KO cells compared to WT cells. To investigate whether the cause for these differences are changes in transcript levels, we performed northern blotting with consecutive probing of mitochondrial transcripts (Fig. 2a). For quantification of transcript levels, loading differences were corrected by normalization to 16S levels (Fig. 2b). This approach gave the best reproducibility and is reasonable since our results showed no effect on mitoribosome levels upon loss of mtRF1 (Supplementary Fig. 1c, d). We observed a clearly significant reduction (~50%) in the transcript levels of COX1 in both KO cell lines. In contrast, we did not detect any significant change for ND6 transcripts, although here smaller changes might not have been detected due to the lower binding efficiency of the ND6 probe.

Next, we investigated whether the decrease in COX1 transcript levels results from degradation due to ribosome stalling on COX1 transcripts upon loss of mtRF1. To address this point, we measured the transcripts levels in both WT and *mtRF1* KO cells before and after inhibition of mitochondrial translation by chloramphenicol (CAM) treatment for 48 h. Inhibition of mitochondrial translation was expected to prevent de novo formation of stalled mitoribosomes at 3' ends of COX1 transcript. While CAM treatment had varied effects on mitochondrial transcripts, we observed a specific increase of COX1 transcript levels in *mtRF1* KO cells upon CAM treatment (Fig. 2c, d and Supplementary Fig. 3a). This implies that degradation of COX1 transcripts upon loss of mtRF1 can, at least to some extent, be rescued by a general inhibition of mitochondrial translation.

Considering the different effects of mtRF1 depletion on COX1 and ND6 transcript levels, we subsequently examined mitochondrial de novo protein synthesis by [${}^{35}$S]-labeling (Fig. 2e, f) and protein steady-state levels by western blotting (Fig. 2g, h). In agreement with our previous observations, *mtRF1* KO cells displayed significantly reduced de novo synthesis of COX1—an effect which is likely directly caused by lower COX1 transcript levels. We did not observe any difference in de novo synthesis of other mitochondrial proteins between WT and *mtRF1* KO cells. Due to the weak translation of ND6, we were unable to reach a clear conclusion regarding the effect of *mtRF1* knockout on de novo ND6 protein synthesis. COX1 protein steady-state levels were decreased even more severely than de novo COX1 synthesis in *mtRF1* KO cells, suggesting significant targeting of newly made COX1 protein for degradation. Detection of ND6 protein steady-state levels by

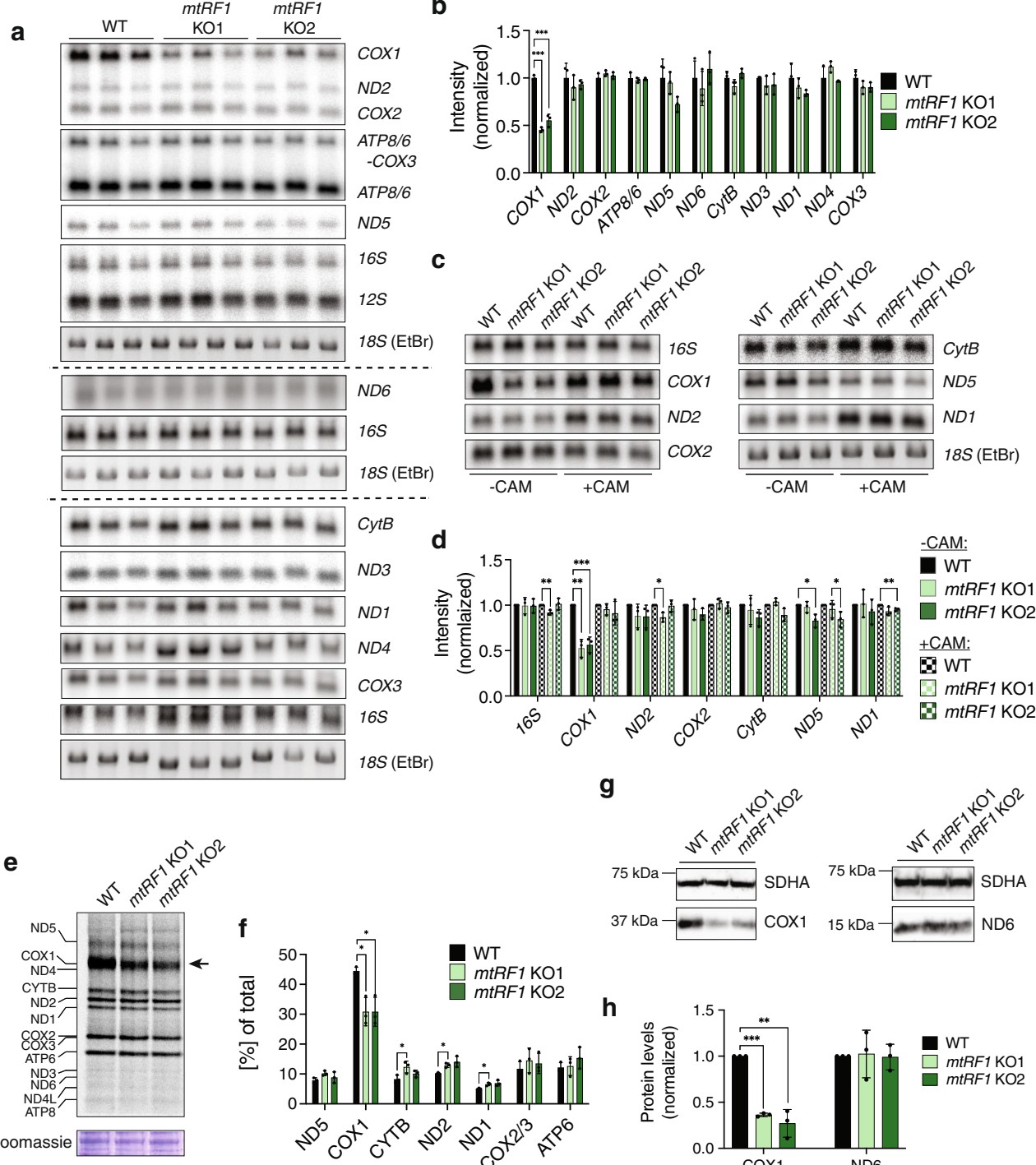

**Fig. 2 | Loss of mtRF1 impairs COX1 mRNA and protein levels. a** Mitochondrial transcript levels of *mtRF1* KO cells compared to WT cells analyzed by northern blotting. Each sample is represented by three biological replicates. Several membranes were used to analyze the whole set of mitochondrial transcripts. Separate membranes are indicated by a dotted line. Loading was assessed by ethidium bromide (EtBr) staining of the gel (18S rRNA band) and 16S probing. **b** Quantification of **a** normalized to transcript levels in WT cells. To account for the difference in loading, all band intensities were first normalized to 16S levels. Means and SD of $n = 3$ biological replicates. Unpaired two-tailed t-test (*$P < 0.05$; **$P < 0.01$; ***$P < 0.001$). **c** Effect of chloramphenicol (CAM) treatment on mitochondrial transcript levels of WT and *mtRF1* KO cells analyzed by northern blotting. A representative blot of three independent experiments is depicted. **d** Quantification of **c** normalized to transcript levels in WT cells +/– CAM treatment, respectively. To

account for the difference in loading, all band intensities were first normalized to 18 S rRNA levels (EtBr). Means and SD of $n = 3$ independent experiments. Unpaired two-tailed t-test (*$P < 0.05$; **$P < 0.01$; ***$P < 0.001$). **e** De novo synthesis of mitochondrial proteins in WT and *mtRF1* KO cells analyzed by [$^{35}$S]-labeling. The arrow highlights differences between WT and *mtRF1* KO cells. Loading was assessed by Coomassie staining. A representative image of three independent experiments is depicted. **f** Quantification of **e** normalized to the total signal of the respective sample. Only clearly visible and well-separated bands were included in the analysis. Means and SD of $n = 3$ independent experiments. Paired two-tailed t-test (*$P < 0.05$; **$P < 0.01$; ***$P < 0.001$). **g** COX1 and ND6 protein levels analyzed by western blotting. Mitochondrial lysates were loaded. Loading was assessed by SDHA detection. **h** Quantification of **g** normalized to protein levels in WT cells. Means and SD of $n = 3$ biological replicates. Unpaired two-tailed t-test (*$P < 0.05$; **$P < 0.01$; ***$P < 0.001$).

western blotting was challenging due to its low abundance, small size, and high hydrophobicity. Several antibodies tested during this study revealed unspecific bands. To assure the correct assignment of ND6, we treated cells with CAM for 48 h, which results in loss of mitochondrially translated proteins. We isolated mitochondria from these cells and assessed ND6 levels by western blotting (Supplementary Fig. 3b). Only the band at 15 kDa disappeared after chloramphenicol treatment and could therefore be assigned to ND6.

In contrast to COX1 protein steady-state levels, ND6 protein levels were not affected by loss of mtRF1. This is consistent with the observation that *ND6* transcript levels are not significantly changed upon mtRF1 depletion.

The fact that we did not observe the loss of ND6 and only partial loss of COX1 protein steady-state levels in the absence of mtRF1 suggests that alternative release mechanisms occur in these cells. Two potential ribosome rescue mechanisms have been described in mitochondria involving the release factors ICT1 and C12orf65 (together with C6orf203)[9,10,15,16,22]. To investigate if induction of mitoribosome stalling in *mtRF1* KO cells results in an upregulation of these proteins, we quantified protein steady-state levels by western blotting. Neither of the two *mtRF1* KO clones showed a significant change of ICT1, C12orf65, or C6orf203 protein levels (Supplementary Fig. 4a, b). To test if loss of these factors impacts COX1 or ND6 translation in *mtRF1* KO cells, we performed knockdown experiments in HEK WT or *mtRF1* KO cells and assessed de novo mitochondrial translation by [³⁵S]-labeling. Here, we focused on C12orf65 and C6orf203 as ICT1 is not only a potential rescue factor but also an integral part of the mitoribosome[22]. Cells were transfected with C12orf65-, C6orf203-, or non-targeted control siRNA and the efficiency of downregulation was assessed by western blotting 72 h after transfection (Supplementary Fig. 4c, d). De novo translation of mitochondrially encoded proteins was not strongly affected by downregulations of C12orf65 and C6orf203 (Supplementary Fig. 4c, d). We observed a minor decrease of de novo translated COX1, which was, however, not dependent on the presence of mtRF1. Therefore, it might reflect a general decrease of translation, which has been reported for loss of functional C12orf65[23] and C6orf203[24]. The quality of the radiographs from C12orf65 knockdown experiments was good enough to quantify de novo synthesized ND6 levels. In agreement with protein steady-state levels, de novo synthesis of ND6 was not affected by the loss of mtRF1. Moreover, it was not affected by the downregulation of C12orf65. Even though these data do not provide evidence for the role of ICT1, C12orf65, and C6orf203 in rescuing stalled mitoribosomes in the absence of mtRF1, their involvement cannot be excluded. While this represents an interesting research topic, it was not pursued further at this point since it was not the major purpose of this study.

In conclusion, our results provide evidence for specific degradation of *COX1* transcripts upon stalling of mitoribosomes at the AGA stop codon and consequent reduction of de novo synthesis of COX1 protein. This mechanism is specific for *COX1* and does not occur for *ND6* upon stalling at the AGG stop codon.

## Loss of mtRF1 triggers distinct effects on OXPHOS protein levels and activity

To investigate the downstream consequences of mitoribosome stalling at AGA/AGG stop codons, we analyzed OXPHOS protein steady-state levels by mass-spectrometry and immunoblotting in WT and *mtRF1* KO cells. Mass-spectrometry data revealed a general decrease of complex IV protein steady-state levels in KO compared to WT cells, while no such decrease was observed for complex I (Fig. 3a and Supplementary Table 4). This result was confirmed by immunoblotting, where we observed a decrease in COX2 but not NDUFB8 (Fig. 3b). COX1 is the first protein to be assembled into complex IV. Therefore, mitoribosome stalling on the *COX1* transcript and the subsequent decrease in incorporation of COX1 into complex IV may impact the

integration of other components of complex IV leading to their degradation. Indeed, it has been previously shown that the absence of COX1 results in depletion of COX2 and COX3 protein levels[25]. Similarly, ND6 is important for complex I integrity and loss of functional ND6 is associated with complex I defiency[26,27]. The fact that complex I levels are not affected by the absence of mtRF1 suggests that ND6 protein is correctly incorporated into complex I despite mitoribosome stalling on the *ND6* transcript and a potential delay of ND6 protein release. Together, these findings further support our observation that ribosome stalling on *COX1* transcript upon loss of mtRF1 has a different effect than stalling on the *ND6* transcript.

Next, we analyzed the impact of loss of mtRF1 on the activity of mitochondrial complexes. Thus, we performed an OXPHOS activity assay on isolated mitochondria (Fig. 3c) as well as BN-PAGE with consecutive in-gel activity and immunoblotting (Fig. 3d, e). As expected, the general decrease in complex IV protein levels also resulted in a significant decrease in complex IV activity (~50%) (Fig. 3c). In addition, we observed a decrease of COX1, COX2, and COX5a proteins in fully assembled complexes in KO cells relative to WT cells (Fig. 3e). The steady-state levels of complex IV assembly factors were only mildly affected by loss of mtRF1 (Supplementary Fig. 5b and Supplementary Table 5). However, we observed a consistent downregulation of the assembly factor PET117 in both KO cell lines, which was suggested to be part of a later COX1 assembly intermediate[28,29]. In contrast, no major effects on complex I activity (Fig. 3c, d) nor levels of assembly factors (Supplementary Fig. 5b and Supplementary Table 5) were observed, further supporting the correct incorporation of ND6 protein into complex I in the absence of mtRF1. Other OXPHOS complexes were not majorly affected by loss of mtRF1 (Fig. 3c).

In summary, our data indicate that release of COX1 from mitoribosomes is crucial for complex IV activity, while impaired release of ND6 has neglectable effects on complex I level and activity.

## An intact GGQ motif is essential for mtRF1 function

The single sequence motif GGQ is highly conserved in all eubacterial, archaebacterial, and eukaryotic release factors. It is located within the PTH domain, which is placed in the peptidyl transferase center of the LSU during peptidyl-tRNA hydrolysis. Mutations within the GGQ motif, such as the substitution of two glycine residues to alanines (AAQ), are associated with loss of release activity[30]. To investigate the importance of the release activity of mtRF1, we generated mtRF1 WT and AAQ (G311A and G312A) rescue cell lines. Since ND6 levels were not affected upon loss of mtRF1, our analysis of the rescue effect focused on COX1 mRNA and protein levels. First, we analyzed transcript levels in both rescue cell lines by northern blotting (Fig. 4a, b). While *COX1* transcript levels were efficiently rescued by introducing mtRF1 WT into *mtRF1* KO cells, no rescue was observed upon the introduction of mtRF1 AAQ. Next, we studied de novo mitochondrial protein synthesis by [³⁵S]-labeling in the generated rescue cell lines (Fig. 4c). MtRF1 WT was able to rescue decreased levels of de novo synthesized COX1, while we observed no rescue upon expression of mtRF1 AAQ. Finally, we assessed protein steady-state levels in both rescue cell lines (Fig. 4d). In accordance with our findings, expression of *mtRF1* WT restored COX1 and COX2 levels, whereas expression of mutant *mtRF1* AAQ reflected the *mtRF1* KO phenotype.

Taken together, our data show that an intact GGQ domain is important for mtRF1 function, confirming that the downstream effects of mtRF1 depletion (i.e., reduction of COX1 transcript and protein levels) were directly caused by the loss of its function as a release factor.

## In vitro, mtRF1 specifically terminates translation at the non-canonical stop codons AGA and AGG

While the release activity of the canonical release factor mtRF1a has been studied biochemically and structurally, the release activity of

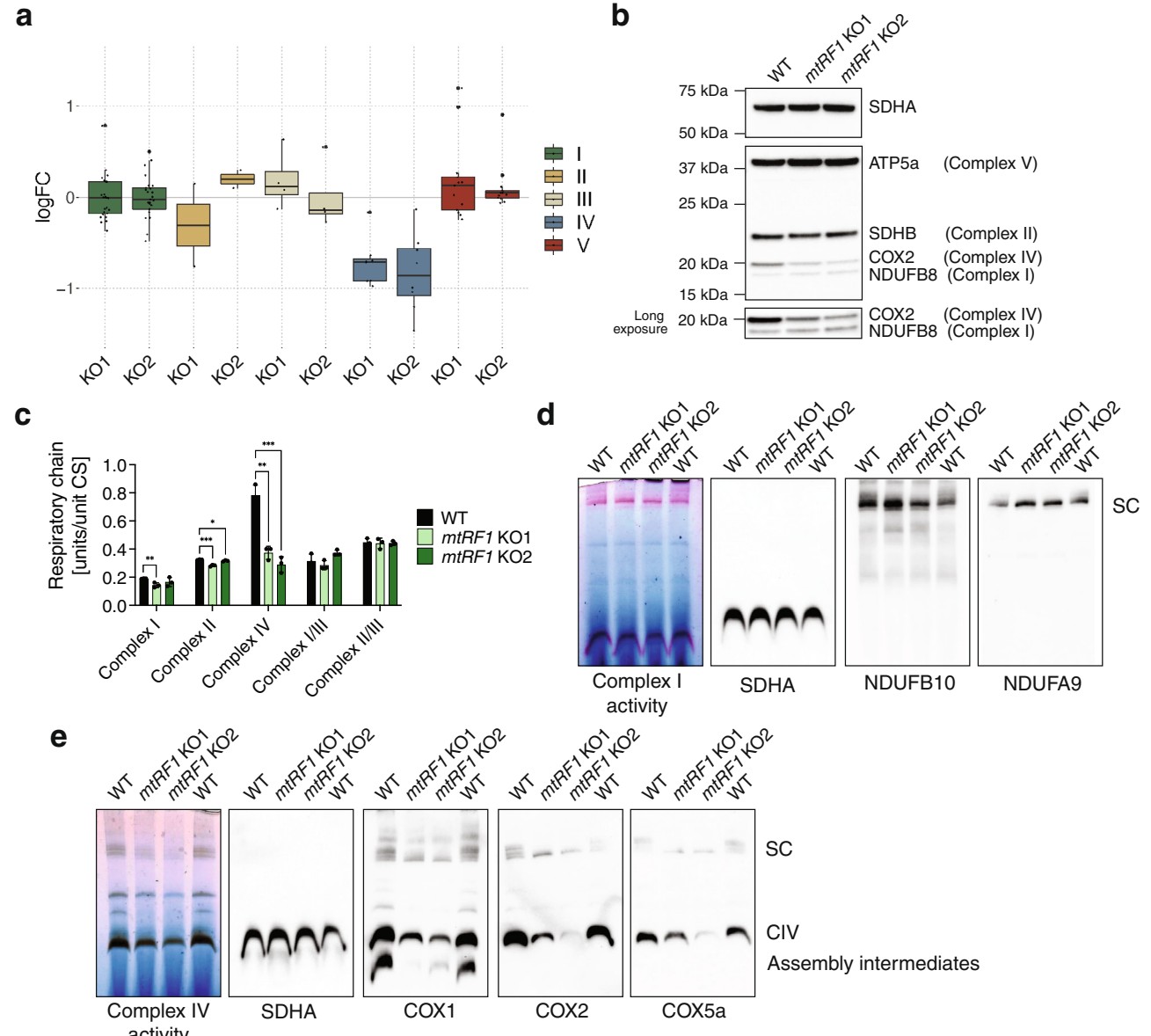

**Fig. 3 | *mtRF1* KO cells exhibit complex IV defect. a** Quantitative mass-spectrometry analysis of OXPHOS complex proteins from HEK *mtRF1* KO cells compared to HEK WT cells. Due to the high hydrophobicity of OXPHOS complex proteins, only a subfraction could be detected in the mass-spectrometer and analyzed. Log2(fold change) (logFC) of *mtRF1* KO1 and *mtRF1* KO2 vs. WT samples is plotted (values are listed in Supplementary Table 4). Each dot represents one protein from the OXPHOS complexes. Peptides were produced from mitochondrial lysates. $n = 3$ independent experiments. Summary of boxplots is depicted in Supplementary Fig. 5a. **b** OXPHOS protein levels analyzed by western blotting. Mitochondrial lysates were loaded. Membranes were probed with the OXPHOS antibody cocktail. Loading was checked by SDHA detection. **c** OXPHOS activity assay. Citrate synthase (CS) activity was used as mitochondrial marker. Means and SD of $n = 3$ independent experiments. Unpaired two-tailed t-test (\*$P < 0.05$; \*\*$P < 0.01$; \*\*\*$P < 0.001$). **d** BN-PAGE and in-gel activity for complex I. The integrity of complex I was analyzed by western blotting (antibodies against NDUFB10 and NDUFA9). Loading was assessed by the detection of SDHA. $n = 1$ experiment. **e** BN-PAGE and in-gel activity for complex IV. The integrity of complex IV was analyzed by western blotting (antibodies against COX1, COX2, and COX5a). Loading was assessed by the detection of SDHA. SC = supercomplexes. $n = 1$ experiment.

mtRF1 remains to be shown by in vitro studies. Mechanistic studies of mitochondrial translation termination are still challenging due to the lack of an efficient in vitro translation system. Recently, a first attempt of reconstituting mitochondrial translation has been published[31], but the low yield of polypeptide synthesis and the high spontaneous peptide release at the stop codons makes it unsuitable for mechanistic studies. Therefore, we developed an in vitro mitochondrial translation system that allows reconstitution of sufficient amounts of post-translocational mitoribosomal complexes (POST) containing a peptidyl-tRNA in the P-site and a stop codon in the A-site. Our in vitro translation system included mitoribosomes isolated from liver mitochondria (Supplementary Fig. 6a), leaderless mRNA (Supplementary

Table 1), human mitochondrial initiation factors mtIF2 and mtIF3, human mitochondrial mtEFG1, and *E. coli* EF-Tu/EF-Ts. Because it is currently challenging to purify and aminoacylate human mitochondrial tRNAs in sufficient amounts required for this study, we used bacterial (fM-tRNA^fM, Q-tRNA^Q) and yeast (F-tRNA^F, R-tRNA^R) tRNAs. Although the mitochondrial counterparts of these tRNAs possess mitochondria-specific post-transcriptional modifications, they maintain a canonical cloverleaf structure[32]. Therefore, such a substitution is likely to be well tolerated in a functional sense. We used co-sedimentation experiments to quantify the synthesis of dipeptide (fMF) and tripeptide (fMFR) and found that ~30% of mitoribosomes are active in translation (Fig. 5a). In addition, we confirmed the identity of

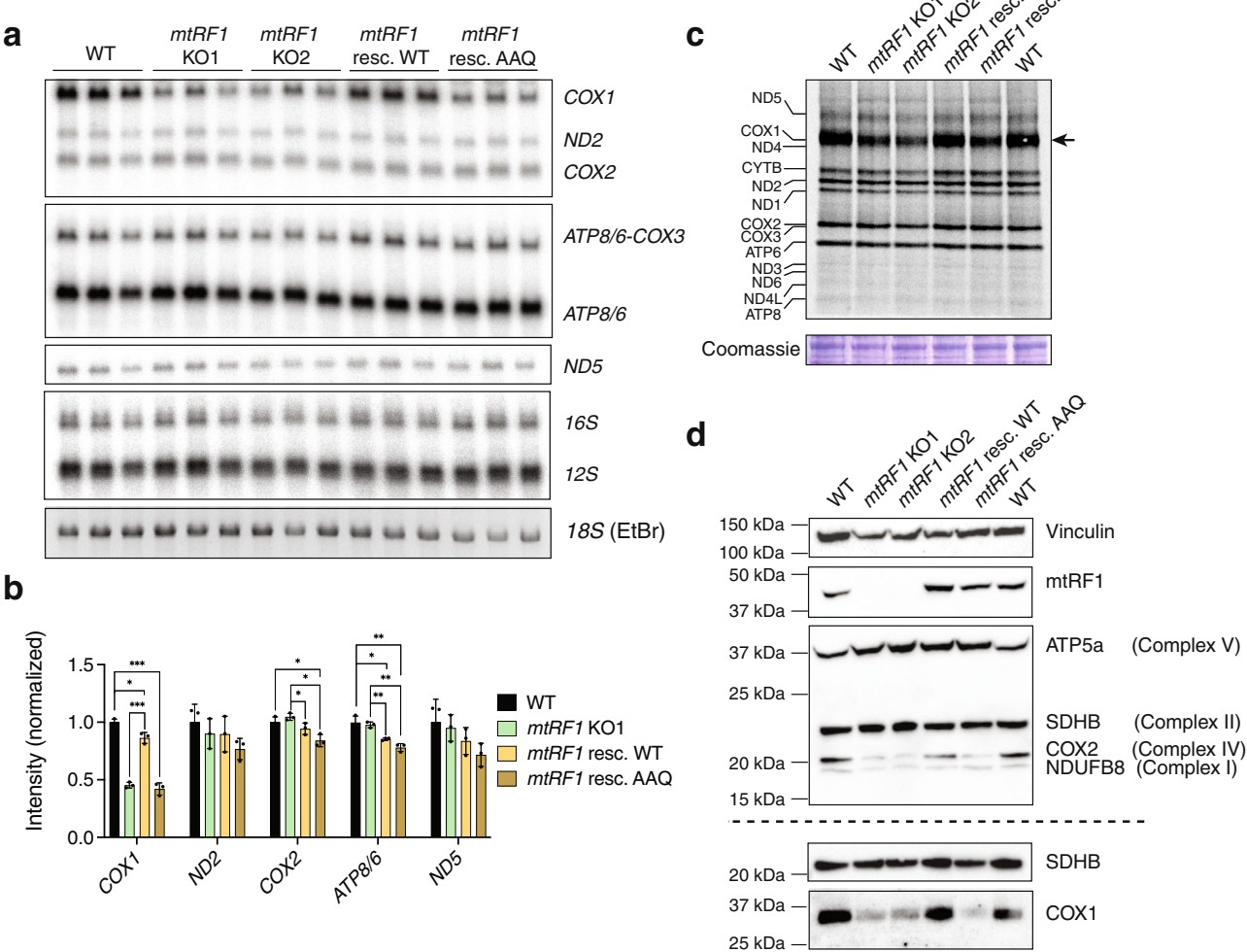

**Fig. 4 | GGQ motif of mtRF1 is essential for its function. a** Mitochondrial transcript levels of mtRF1 rescue cells (WT and AAQ mutant) compared to *mtRF1* KO and WT cells analyzed by northern blotting (the same membrane as in Fig. 2a is shown). Each sample is represented by three biological replicates. Several transcripts were probed on one membrane. Loading was assessed by ethidium bromide (EtBr) staining of the gel (18S rRNA band) and 16S probing. **b** Quantification of **a** normalized to transcript levels in WT cells. To account for different loading, all band intensities were first normalized to 16S levels. Means and SD of $n = 3$ biological replicates. Unpaired two-tailed t-test of rescue cells compared to WT and *mtRF1* KO1 cells. (*$P < 0.05$; **$P < 0.01$; ***$P < 0.001$). **c** De novo synthesis of mitochondrial

proteins in WT, *mtRF1* KO and mtRF1 rescue cells analyzed by [$^{35}$S]-labeling (the same gel as in Fig. 2e is shown). The arrow highlights differences of COX1 protein synthesis between WT, *mtRF1* KO and mtRF1 rescue cells. Loading was assessed by Coomassie staining. $n = 1$ experiment. **d** OXPHOS protein levels analyzed by western blotting. 100 µg of whole-cell lysates were loaded. Membranes were probed with the OXPHOS antibody cocktail (upper panel) and anti-COX1 antibody (lower panel). The expression of mtRF1 in rescue cells was confirmed by the anti-mtRF1 antibody. Loading was assessed by Vinculin/SDHB detection. A representative blot of $n = 3$ independent experiments is depicted.

the translated tripeptide by thin-layer electrophoresis (Supplementary Fig. 6b). To exclude a contamination of mitoribosomes with cytosolic ribosomes, we determined the amount of synthesized tripeptide in presence of 0.5 mM fusidic acid. As expected, fusidic acid almost completely inhibited pentapeptide synthesis in a eukaryotic cytosolic in vitro translation system, but did not affect mitoribosome translation (Fig. 5a).

To investigate the in vitro functions of mitochondrial translation termination factors, we used post-translocation complexes (POST2) programmed with the mRNAs described in Supplementary Table 1, containing fMQ-tRNA$^Q$ in the P-site and either the canonical stop codon UAG or one of the noncanonical termination codons (AGA or AGG) in the A-site (Fig. 5b). After addition of either mtRF1 or mtRF1a and incubation at 37 °C for 5 min, we determined the percentage of released peptide by co-sedimentation (Fig. 5b, c). To correct for spontaneous peptide release, we measured the amount of released peptide in the absence of release factors and used this number as

background (10–15 % of the total peptide was spontaneously released in our conditions). As previously reported, mtRF1a catalyzed peptide release at the standard stop codon UAG (-70% release activity) but not at the non-canonical stop codons AGA/AGG (Fig. 5c)[7,8,10]. In contrast, we observed a significant release activity of mtRF1 at the non-canonical stop codons AGA (37%) and AGG (24.5%), but no activity at the standard stop codon UAG.

In conclusion, our in vitro experiments demonstrate that mtRF1 has a specific release activity at the non-canonical stop codons AGA/AGG and not at the standard UAG stop codon, supporting our analysis of mtRF1 KO cells.

## Discussion
The role of mtRF1 has been debated for a long time[2,3,7–12,17,18], yet none of the previous studies provided direct evidence for its function. Here, we show that cells lacking mtRF1 display stalling of mitoribosomes at the non-canonical stop codons AGA and AGG. Furthermore, we

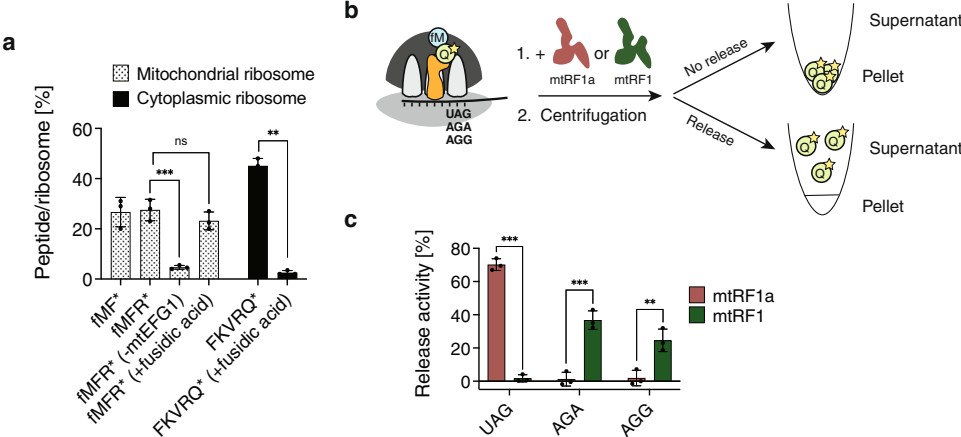

**Fig. 5 | mtRF1 specifically acts on non-canonical stop codons AGA/AGG. a** In vitro translation assay using mitochondrial or cytosolic ribosomes. Up to five amino-acylated tRNAs were added to the translation mixture. Elongation was assessed by measuring incorporation of radioactively labeled amino acids (*). Specificity of mitoribosomal translation was tested by the absence of mtEFG1 or by addition of fusidic acid. Means and SD of $n = 3$ independent experiments. Unpaired two-tailed t-test (*$P < 0.05$; **$P < 0.01$; ***$P < 0.001$; ns = not significant). Only relevant statistics are highlighted. **b** Schematic representation of the in vitro mitochondrial translation termination assay. Post-translocation complexes (POST2) containing fMQ-tRNA$^{Q^*}$ in the P-site and either canonical stop codon UAG or non-canonical stop codons AGA/AGG in the A-site were prepared. MtRF1a/mtRF1-dependent release of dipeptide was assayed by co-sedimentation. Radioactive glutamine was used for detection. **c** In vitro release activity of mtRF1a and mtRF1 at UAG/AGA/AGG codons. Means and SD of $n = 3$ independent experiments. Unpaired two-tailed t-test (*$P < 0.05$; **$P < 0.01$; ***$P < 0.001$).

developed an in vitro mitochondrial translation assay that allowed us to demonstrate the specific peptide release activity of mtRF1 on mitoribosomes programmed with AGA and AGG codons in the A-site. Our study thus provides an indication that mtRF1 can recognize AGA/AGG triplets as stop codons, leading to the release of full-length COX1 and ND6 proteins from the ribosome.

Previous studies utilizing in vitro translation assays failed to detect binding or release activity of mtRF1 on AGA/AGG codons[7–10]. Here, we optimized previously published translation assays to analyze the specific activity of mtRF1 on all mitochondrial stop codons. While previous studies employed initiation complexes, we used post-translocational complexes containing fMQ-tRNA$^Q$ in the ribosomal P-site and stop codons in the A-site. The identity of the peptidyl-tRNA in the ribosomal P-site has been shown to influence the rates of peptide release catalyzed by RF1 and RF2 in the bacterial system[33]. Consequently, optimizing the amino acid context might further improve the release activity of mtRF1 over the 40–50% fMQ released in our current assay. Methylation of the GGQ motif of recombinant mtRF1 might further improve its release activity. This modification is highly conserved among species and has been shown to impact release activity under certain conditions[33]. The methyltransferase methylating all mitochondrial release factors including mtRF1 has been identified[34,35], however, its role requires further characterization.

The hypothesis of a −1 frameshift at non-canonical stops codons which would place the standard stop codon UAG in the A-site recognized by mtRF1a has been proposed to obviate the need for an mtRF1-dependent termination mechanism[2]. However, most vertebrate species having AGA/AGG at the end of mtORFs cannot place a canonical stop codon by −1 frameshifting in their A-site[3]. To the best of our knowledge, all species having AGA/AGG at the end of mtORFs also have *MTRF1*, supporting a parallel evolution of the stop codon and the matching release factor. Interestingly, a few species (e.g., Rattus norvegicus, Mus musculus, Platichthys flesus, and Danio rerio) have *MTRF1* but lack AGA/AGG codons in their mitochondrial genome[18]. A similar scenario has been observed for other species, in which RF2 is retained despite the loss of the stop codon UGA, which it recognizes[3,36]. The preservation of release factors during evolution has been suggested to prevent the reassignment of codons[36]. Moreover, having an additional release factor provides further control during the translation process as ribosome frameshifting errors can be terminated at multiple codons. Consequently, phylogenetic analysis favors our hypothesis of mtRF1 being a canonical release factor, questioning whether −1 frameshifting events are required to terminate translation. In Temperley et al., a bacterial endoribonuclease, which specifically cleaves mRNA in ribosomal A-site, was targeted to mitochondria, giving indirect information about mitoribosome positions. The introduction of a bacterial endonuclease with a strong preference for the UAG codon over AGA/AGG codons[37], might have favored cleavage at UAG codons instead of AGA/AGG codons and thus generated potential artefacts.

Interestingly, we observe at least partial integration of COX1 and ND6 proteins into OXPHOS complexes in the absence of mtRF1, suggesting the presence of additional release mechanisms, which resolve stalled mitoribosomes and result in the release of COX1 and ND6 proteins. Two rescue mechanisms have been described in mitochondria so far involving the release factors ICT1 and C12orf65. Besides being an integral part of the mitoribosome, ICT1 was shown to possess peptide release activity on nonstop ribosomes[9,10] and ribosomes stalled on mRNAs containing a maximum of 14 nucleotides past the A-site[16]. As *COX1* and *ND6* transcripts include 3′-UTRs >14 nucleotides in length, ICT1 would have no release activity on these transcripts. However, it is possible that mRNA on stalled mitoribosomes is targeted for cleavage by a currently unknown mechanism, giving access to ICT1. In bacteria as well as in the eukaryotic cytosol, it has been recently shown that ribosome stalling within ORFs leads to ribosome collisions, which trigger mRNA cleavage and ribosome rescue[38,39]. Interestingly, our ribosome profiling data indicate the presence of additional stalling events upstream of AGA/AGG codons (Fig. 1f and Supplementary Fig. 2b), potentially resembling collided mitoribosomes. Thus, a similar rescue mechanism might occur in mitochondria. The exact function of the rescue factor C12orf65 is currently not well understood. A recent cryo-EM study of mitoribosomes isolated from *PDE12* KO cells identified C12orf65 together with C6orf203 in an mtLSU complex containing peptidyl-tRNA in P-site[15]. As KO of *PDE12* results in depletion of Lys-tRNA$^{Lys}$ [40], C12orf65 was suggested to be involved in the rescue of no-go decay complexes. It remains unclear how stalled mitoribosomes are recognized and split, and what happens to the released mRNA. As stalling due to the lack of tRNA presents similar features to stalling due to the lack of release factors, the same rescue mechanisms might be involved. However, our preliminary results do not provide evidence for

C12orf65/ C6orf203 involvement in the mitoribosome rescue upon loss of mtRF1 (Supplementary Fig. 4), highlighting a need for further studies on mitochondrial translation quality control mechanisms.

In our *mtRF1* KO model, *COX1* transcript levels were significantly decreased, while *ND6* transcript levels were not substantially affected. One potential reason for this difference might lay in the structure of the two transcripts. While *COX1* mRNA harbors a relatively short 3' UTR and is post-transcriptionally modified with a 3'-polyA tail, *ND6* possesses a long 3'UTR and is the only transcript in human mitochondria lacking a polyA tail. The function of polyA tails in mitochondrial mRNA remains elusive as complete depletion of polyA tails was associated with varying effects on transcript levels[41]. Interestingly, KO of the mRNA-binding protein LRPPRC in mice resulted in shortening of all polyA tails and a general decrease in all transcripts except *ND6*[42]. Consequently, a length-dependent function of polyA tails was suggested, which is a known regulatory mechanism for cytosolic transcripts[43]. Consistent with this, a microdeletion within *ATP6* ORF, which removed the stop codon and likely induced stalling of mitoribosomes downstream of the stop codon, resulted in shortening of polyA tails on *ATP8/ATP6* transcripts and a reduction of *ATP8/6* transcript level[44]. In this work, the shortening of polyA tails and the associated reduction of transcript levels was reversed by inhibiting mitochondrial translation by thiamphenicol (TAM) treatment—a phenomenon similar to our observations on *COX1* transcript levels upon CAM treatment. Thus, it can be speculated that stalling of mitoribosomes specifically targets mRNAs for degradation in a polyA-dependent manner, which can be prevented by inhibiting de novo occurrence of the stalling events.

While stalling of mitoribosomes at the end of *COX1* transcripts resulted in reduction of complex IV protein levels and activity, stalling at the end of *ND6* had negligible effects on complex I. One of the reasons for these differences can be the reduction in transcript levels of *COX1* but not *ND6*. However, as COX1 protein steady-state levels were more severely reduced than COX1 de novo protein synthesis, it is likely that other factors contributed to these differences; for example, the distinct incorporation of the two proteins into OXPHOS complexes. COX1 is the first protein to be incorporated into complex IV and is a core subunit. Loss of COX1 is linked to a total loss of complex IV holoenzyme and a substantial decrease of several complex IV proteins, including COX2 and COX3[25]. Thus, delayed release of COX1 due to mitoribosome stalling might affect the initial assembly of complex IV, preventing stable incorporation of early complex IV proteins into the inner mitochondrial membrane. This hypothesis is in agreement with our data showing a decrease in the steady-state levels of early complex IV proteins (e.g., COX2, COX5a) and assembly factors (e.g., PET117) upon loss of mtRF1. Moreover, it has been previously shown that truncated COX1 subunits are unstable and constitutively targeted for degradation[45]. In contrast to COX1, ND6 is incorporated at a later stage of complex I assembly. Therefore, the assembly intermediates might be stabilized by several complex I proteins and assembly factors, tolerating delayed release of ND6 protein. Functional ND6 was shown to be vital for complex I activity[26,27] and patients harboring mutations within the *ND6* gene develop Leber's hereditary optic neuropathy (LHON) due to complex I failure[46–48]. Therefore, even though mitoribosome stalling on *ND6* transcripts in *mtRF1* KO cells does not seem to impact complex I function, its eventual release (possibly with the help of rescue mechanism) is essential. Even though we did not observe an impact on complex I upon deletion of mtRF1, this might change under certain stress conditions. In addition, it would be interesting to see if the effect of mitoribosome stalling on *COX1* and *ND6* transcripts could be tissue-specific and possibly heightened by various pathophysiological conditions. As our studies are performed in HEK cell lines only, further in vivo analysis are needed to address this issue.

In conclusion, our data show that human mitochondria require two release factors to terminate the translation of all 13 mitochondrially encoded proteins: mtRF1a for the standard stop codons UAA and UAG and mtRF1 for the non-canonical stop codons AGA and AGG. In addition, we demonstrate that mitochondrial protein synthesis, peptide release, and integration of the synthesized proteins into OXPHOS complexes are tightly controlled processes, which can have severe downstream effects once impaired. Importantly, we reveal the existence of transcript-specific downstream mechanisms triggered by stalling of mitoribosomes. Future studies will elucidate the ribosome rescue pathways in human mitochondria, providing details on how stalling events are recognized, the factors involved in this process, and how the translated mRNA is stabilized or targeted for degradation.

## Methods

### Experimental model

Flp-In T-Rex human embryonic kidney 293 (HEK293T) cell line (Thermo Fisher, RRID: CVCL_U427) was used to generate *mtRF1* KO and mtRF1 rescue cell lines. All cell lines were maintained in DMEM (high glucose, GlutaMAX, sodium pyruvate) (Thermo Fisher) supplemented with 10% (v/v) fetal bovine serum (FBS) (Thermo Fisher), 1× Penicillin/ Streptomycin (Thermo Fisher) and 50 μg/ml uridine at 37 °C and 5% $CO_2$ unless stated otherwise. For maintenance of rescue cell lines, medium was supplemented with 100 μg/ml blasticidin (Thermo Fisher) and 100 μg/ml hygromycin B (Thermo Fisher).

### Generation of *mtRF1* KO cell lines

*MtRF1* KO cells were generated from HEK293T cell line by CRISPR/ Cas9 system according to Ran et al.[49]. Two single guide RNAs (sgRNAs) targeting exon 2 in *MTRF1* where designed such as they would generate an out-of-frame deletion (guide site 1: TGTTAAGTAAGAATTGGTCC / guide site 2: GAACAGAAGGCATGCTGAGT). SgRNAs were cloned into pSpCas9(BB)−2A-Puro (PX459) V2.0 vector individually and transfected together into HEK293T cell line using Lipofectamine 3000 (Thermo Fisher) according to manufacturer's instructions. The next day, transfected cells were selected by 1.5 μg/ml puromycin treatment for 48 h. Consecutively, clonal cell lines were isolated by dilution in 96-well plates. *MtRF1* KO cell lines were first screened by PCR using primers targeting the region around the CRISPR/Cas9 cut site (GCT GCAAGTTTTTAGACAAAACAGG, TTCTTCAATTGCTTGTTCAGTCTCC) (listed in Supplementary Table 1). Selected clones were finally verified by immunoblotting of mtRF1 and Sanger sequencing.

### Generation of mtRF1 rescue cell lines

For cloning procedures, One Shot™ TOP10 Chemically Competent *E. coli* cells (Thermo Fisher) were used. Cloning into pcDNA5/FRT/TO vector was confirmed via Sanger sequencing with CMV and BGH primers (listed in Supplementary Table 1).

MtRF1 ORF (acc. no NM_004294), provided by GenScript, was cloned with C-terminal FLAG tag into pcDNA5/FRT/TO vector (BamHI and XhoI sites) using BamHI-mtRF1-FLAG-F and mtRF1-FLAG-XhoI-R primers (listed in Supplementary Table 1). C-terminal FLAG tag was then removed using mtRF1-noFLAG-F and mtRF1-noFLAG-R primers (listed in Supplementary Table 1). AAQ mutant variant of mtRF1, carrying Gly311Ala and Gly312Ala substitutions, was obtained by mutagenesis using mtRF1-AAQ-mut-F and mtRF1-AAQ-mut-R primers (listed in Supplementary Table 1).

MtRF1 and mtRF1 AAQ mutant were transfected together with pOG44 Flp-Recombinase Expression vector into HEK293T *mtRF1* KO cells using Lipofectamine 3000 (Thermo Fisher) according to manufacturer's instructions. This allowed stable and inducible re-expression of *mtRF1*. 48 h after transfection, cells were selected by supplementing media with 100 μg/ml blasticidin (Thermo Fisher) and 100 μg/ml hygromycin B (Thermo Fisher). After 2−3 weeks, colonies were picked and cultured as single clonal populations. Expression of *mtRF1* was induced by 10 ng/ml doxycycline treatment for 48 h and checked by immunoblotting of mtRF1.

## Growth curve

$5 \times 10^4$ cells were plated in 6-well plates and cultured in standard DMEM medium for 24 h. Consecutively, medium was replaced by galactose medium (10 mM Galactose, 10% (v/v) FBS, 2 mM GlutaMax, 1× Penicillin/Streptomycin and 1× Sodium Pyruvate) to test mitochondrial function. Cell number was determined every 48 h up to 6 days (post medium change) by EVE Automated Cell Counter (NanoEnTek). As a control, cells were cultured in standard DMEM medium and counted.

## Isolation of mitochondria

All steps were performed on ice or at 4 °C. Cells were harvested at 90% confluency and resuspended in MSE buffer (150 mM D-mannitol, 100 mM Tris pH 7.4, 1 mM EDTA) supplemented with 0.1% (w/v) BSA. Cells were lysed by homogenization (Homgenplus Homogenizer, Schuett-biotec) and mitochondria were isolated by differential centrifugation ($400 \times g$, 10 min and $11,000 \times g$, 10 min). Mitochondrial pellets were resuspended in 200–300 µl MSE buffer (w/o BSA), loaded on a sucrose cushion (1–1.5 M sucrose, 11 ml total, 20 mM Tris pH 7.4, 1 mM EDTA) and centrifuged for 1 h at $106,880 \times g$ (SW41-Ti rotor, Beckman Coulter). Mitochondria, which were located at the interface of both sucrose layers, were collected, and mixed with equal volumes of 10 mM Tris pH 7.4 buffer to lower the sucrose concentration. Finally, mitochondria were pelleted ($11,000 \times g$, 10 min), resuspended in freezing buffer (300 mM trehalose, 10 mM Tris pH 7.4, 10 mM KCl, 1 mM EDTA, 0.1% BSA), shock frozen in liquid nitrogen and stored at −80 °C. Before starting an experiment, mitochondria were thawed on ice, pelleted by centrifugation ($11,000 \times g$, 10 min), and washed using 500 µl MSE buffer. Mitochondria were quantified by Qubit protein quantification assay (Thermo Fisher).

## Quantitative mass-spectrometry

50 µg of mitochondria from HEK WT, *mtRF1* KO1 and *mtRF1* KO2 cells were resuspended in 100 µl 6 M guanidine hydrochloride in 100 mM Tris pH 8.0. For complete denaturation of proteins, samples were sonicated two times for 5 min (10 s on/10 s off). Afterwards, samples were centrifuged at max. speed for 10 min and supernatants were transferred to new tubes. Proteins were reduced by a final DTT concentration of 5 mM for 30 min at 55 °C and consecutively cooled down on ice. Next, proteins were alkylated at RT in the dark for 15 min by adding chloroacetamide in a final concentration of 15 mM. During alkylation samples were quantified by BCA assay. Alkylation was stopped by adding 10 × 50 mM Tris pH 8.0 buffer. Proteins were digested by trypsin (1:50 w/w) ON at 37 °C. Peptides were purified by Pierce Peptide Desalting Spin columns (Thermo Fisher) according to the manufacturer's protocol. Four micrograms of desalted peptides were dried out and reconstituted in 9 µL of 0.1 M TEAB. Tandem mass tag (TMTpro, lot number WC314415, Thermo Fisher) labeling was carried out according to the manufacturer's instruction with the following changes: 0.5 mg of TMTPro reagent was resuspended with 33 µL of anhydrous ACN (labeling scheme is summarized in Supplementary Table 6). Seven microliters of TMTPro reagent in ACN was added to 9 µL of clean peptide in 0.1 M TEAB. The final ACN concentration was 43.75% and the ratio of peptides to TMTPro reagent was 1:20. After 60 min of incubation, the reaction was quenched with 2 µL of 5% hydroxylamine. Labeled peptides were pooled, dried, resuspended in 200 µL of 0.1% formic acid (FA), split into two equal parts, and desalted using home-made STAGE tips[50]. One of the two parts was fractionated on a 1 mm × 150 mm ACQUITY column, packed with 130 Å, 1.7 µm C18 particles (Waters), using an Ultimate 3000 UHPLC (Thermo Fisher). Peptides were separated at a flow of 30 µL/min with an 88 min segmented gradient from 1 to 50% buffer B for 85 min and from 50 to 95% buffer B for 3 min (buffer A: 5% ACN, 10 mM ammonium bicarbonate (ABC); buffer B: 80% ACN, 10 mM ABC). Fractions were collected every three min, and fractions were pooled in

two passes (1 + 17, 2 + 18 ... etc.) and dried in a vacuum centrifuge (Eppendorf). Dried fractions were resuspended in 0.1% formic acid (FA) and separated with an EASY-nLC1200 running a 90 min linear gradient from 6 to 31% buffer B (buffer A: 0.1% FA; buffer B: 0.1% FA, 80% ACN) on a 40 cm, 75 µm internal diameter, unfritted analytical column (CoAnn Technologies, Parts Number: ICT3607508-50-5) packed with 2.7 µm C18 Poroshell media (Agilent, Part Number: 660120-002). The column was operated at 50 °C. Eluting peptides were analyzed on an Orbitrap Fusion Tribrid mass-spectrometer (Thermo Fisher). Peptide precursor m/z measurements (MS1) were carried out at in the Orbitrap at 60,000 resolution, 350 to 1500 m/z range, using an automatic gain control (AGC) target of 4e5. The cycle time was set to 1.5 s. The most intense precursors with charge states 2 to 6 only were selected for collision-induced fragmentation using 35% collision energy. Upon fragmentation, precursors were put on a dynamic exclusion list for 60 s. The corresponding fragmentation patterns (MS2) were acquired in the IonTrap using a "Normal" scan rate, 44 ms injection time, and an AGC target of 1e4. Isobaric TMTPro reporters were excluded. The ten most intense fragments were further isolated using synchronous precursor selection and fragmented by higher-energy C-trap dissociation using 65% collision energy. The corresponding fragmentation spectra (MS3), containing the TMTPro reporter ions, were acquired in the Orbitrap at a resolution of 50000, AGC target of 1e5, and 86 ms maximum injection time. Proteomics data was analyzed using MaxQuant, version 1.6.17.0[51]. Peptide fragmentation spectra were searched against the canonical sequences of the human reference proteome (proteome ID UP000005640, downloaded September 2018 from UniProt). Methionine oxidation and protein N-terminal acetylation were set as variable modifications; cysteine carbamidomethylation was set as fixed modification. The digestion parameters were set to "specific" and "Trypsin/P". Quantification was set to "Reporter ion MS3". The isotope purity correction factors, provided by the manufacturer, were imported and included in the analysis. The minimum number of peptides and razor peptides for protein identification was 1; the minimum number of unique peptides was 0. Protein identification was performed at a peptide spectrum matches and protein false discovery rate of 0.01. The "second peptide" option was on. Differential expression analysis was performed using limma, version 3.34.9[52] in R, version 3.4.3[53].

For analysis of steady state levels of mitoribosomal proteins, logFC values of proteins belonging to the mitochondrial ribosome according to MitoCarta3.0 were plotted (Supplementary Table 3). For analysis of steady state levels of OXPHOS complexes, logFC values of proteins belonging to OXPHOS complexes I–V according to MitoCarta3.0 were plotted (Supplementary Table 4). For analysis of steady state levels of OXPHOS assembly factors, logFC values of proteins belonging to OXPHOS assembly factors according to MitoCarta3.0 were plotted (Supplementary Table 5). Data are represented by box plots.

## Sucrose gradient centrifugation

All steps were performed on ice or at 4 °C. Linear sucrose gradients (10–30% w/v, 2.5 ml total volume) were prepared in 20 mM Tris pH 7.4, 100 mM KCl, 20 mM MgCl$_2$ and 1× Complete EDTA-free protease inhibitor cocktail (Roche) using a Gradient Master (BioComp). 1.25 mg of mitochondria from HEK WT, *mtRF1* KO1 and *mtRF1* KO2 cells were lysed for 20 min in 100 µl lysis buffer (20 mM Tris pH 7.4, 100 mM KCl, 20 mM MgCl$_2$, 1% Triton X-100, 1 × Complete EDTA-free protease inhibitor cocktail (Roche), 0.4 U/µl RNase block RNase inhibitor (Agilent Technologies)). Cell debris were removed by centrifugation ($5000 \times g$, 10 min) and supernatants were loaded on prepared sucrose gradients. Samples were centrifuged at $130,000 \times g$ for 2 h 15 min (TLS55, Beckman Coulter). Afterwards, 100 µl fractions were collected and 10 µl of each fraction were analyzed by immunoblotting using the depicted antibodies.

## Mitoribosome profiling

Mitoribosome profiling was performed according to Pearce et al.[20]. HEK WT and *mtRF1* KO cells were grown to 80% confluency on 15 cm dishes. Medium was discarded and plates were shortly submerged in liquid nitrogen to snap freeze cells. 2x lysis buffer (100 mM Tris pH 7.5, 200 mM NaCl, 40 mM MgCl$_2$, 2 mM DTT, 200 µg/ml chloramphenicol, 200 µg/ml cycloheximide, 2% Triton X-100, 2× Complete EDTA-free protease inhibitor cocktail (Roche), 4000 U/ml TURBO DNase I (Thermo Fisher)) was added dropwise on the plates and lysates were collected using cell scrapers. Lysis was completed by triturating samples 10× with a 20-G needle and cell debris were removed by centrifugation (13,000 × $g$, 20 min, 4 °C). 200 µl of supernatant were subjected to RNase treatment for 30 min at RT (450 U, Ambion RNase I, Thermo Fisher). RNase treatment was stopped by the addition of 10 µl RNase inhibitor (1 U/µl, SUPERase-In, Thermo Fisher), which was followed by a short centrifugation step to remove insoluble material (5000 × $g$, 5 min). Supernatants were loaded on 10–30% sucrose gradients (50 mM Tris pH 7.4, 100 mM NaCl, 20 mM MgCl$_2$, 1 mM DTT, 100 µg/ml chloramphenicol, 100 µg/ml cycloheximide, 1× Complete EDTA-free protease inhibitor cocktail (Roche), 40 U/ml RNase inhibitor (SUPERase-In, Thermo Fisher)) in 11 × 34 mm tubes (Beckman Coulter) and run at 130,000 × $g$ for 2 h 15 min in a TLS-55 rotor (Beckman Coulter). Afterwards, 100 µl fractions were collected, and 10 µl of each were used for western blot analysis. The remaining volumes of fractions 11–16, representing monosome containing fractions, were combined, and further processed for mitoribosome profiling. First, RNA was extracted as mentioned in paragraph 'Northern blotting – RNA extraction'. Isolated RNA was heated at 80 °C for 3 min, put on ice for 1 min, mixed with Gel Loading Buffer II (Thermo Fisher) and loaded onto a 15% Novex TBE-Urea gel (Thermo Fisher). The gel was run in 1x TBE buffer at 100 V for ~2 h. After completion of the run the gel was stained with 1× SYBR Gold Nucleic Acid Gel Stain (Thermo Fisher) in 1× TBE. Nucleic acids were visualized and bands referring from 30 to 40 nt were excised. RNA was extracted from gel slices in 600 µl RNA extraction buffer (300 mM NaOAc pH 5.5, 1 mM EDTA, 0.25 % SDS) rotating at 4 °C ON. The next day, RNA was precipitated by adding 1.8 ml ice-cold EtOH together with 4 µl GlycoBlue Coprecipitant (Thermo Fisher) and subsequent storage at −80 °C ON. Precipitated RNA was pelleted by centrifugation (5000 × $g$, 10 min, 4 °C). Pellet was once washed with 1 ml EtOH, dried for ~5 min and resuspended in 15 µl 10 mM Tris pH 7.5 supplemented with 1 µl RNase inhibitor (SUPERase-In, Thermo Fisher). Samples were heated at 80 °C for 2 min before placing on ice. Next, 3' phosphates were removed by T4 PNK treatment (1 µl T4 PNK (NEB) added) in 1× T4 PNK buffer (NEB) at 37 °C for 2 h. Reaction was stopped by heat inactivation (65 °C, 10 min). RNA was pelleted by addition of 70 µl water, 2 µl GlycoBlue Coprecipitant (Thermo Fisher), 10 µl 1 M NaOAc and 300 µl EtOH and subsequent storage at −80 °C. RNA was washed and dried as described earlier and finally resuspended in 7 µl 10 mM Tris pH 7.5 supplemented with 1 µl RNase inhibitor. RNA libraries were generated using TrueSeq Small RNA Library Prep Kit (Illumina) according to the manufacturer's protocol with some modifications. Preparation was started by adding 1.2 µl adenylated RA3 to dephosphorylated RNA and incubating the mixture at 80 °C for 2 min. Afterwards, ligation was performed by the addition of 2 µl of T4 RNA Ligase 2 (truncated K227Q), 2 µl T4 RNA Ligase 2 buffer and 6 µl PEG8000 (all components from NEB) and incubation at 14 °C ON. RNA was precipitated as described earlier, 20 µl 3 M NaOAc and 600 µl EtOH) and resuspended in 4 µl 10 mM Tris pH 7.5. Ligation products were then purified on a 15% Novex TBE-Urea gel (Thermo Fisher), extracted, and precipitated as described earlier. Next, RNA was resuspended in 13 µl 10 mM Tris pH 7.5 supplemented with 1 µl RNase inhibitor. Then, 2 mM ATP, 2 µl 10× T4 PNK buffer and 2 µl T4 PNK (NEB) were added, and the reaction mixture was incubated for 2 h at 37 °C, followed by heat inactivation (65 °C, 10 min). RNA was precipitated and resuspended in 13 µl 10 mM Tris pH 7.5 supplemented

with 1 µl RNase inhibitor. Thereafter, RNA footprints were ligated with 5' RNA adaptor (RA5, Illumina) by adding 1.2 µl RA5, 2 µl 10× T4 buffer and 2 µl T4 RNA ligase (Promega) and incubating at 14 °C ON. RNA was precipitated and resuspended in 3 µl 10 mM Tris pH 7.5. Reverse transcription was performed using RNA RT primers from TrueSeq Small RNA Library Prep Kit (Illumina) and SuperScript III First-Strand Synthesis System (Thermo Fisher) according to the manufacturer's protocol. Afterwards, 2 µl of RT products were PCR amplified using Phusion High-Fidelity PCR master mix (NEB) and DNA primers from TrueSeq Small RNA Library Prep Kit (Illumina). The PCR products were purified using QIAquick PCR Purification Kit (Qiagen), precipitated as usual and resolved on a 10% Novex non-denaturing TBE gel (Thermo Fisher) using 1× TBE running buffer. PCR products were excised and extracted using DNA extraction buffer (300 mM NaCl, 10 mM Tris pH 8, 1 mM EDTA). Subsequently, PCR products were precipitated and pelleted. Libraries were resuspended in 11 µl 10 mM Tris pH 7.5 and 3 ng were used for sequencing, respectively on the Illumina NextSeq 500. The pooled sequencing data were split by the index barcodes used for each sample during the pooling PCR and converted to the FASTAQ format by bcl2fastq (Illumina). All the FASTAQ files were used as input for the MitoRiboSeq analysis pipeline[54]. The MitoRiboSeq package with minor modifications, which harbored the open-source software and custom Python, and R codes were used for the mtRPFs sequencing data analysis. First, the common 3' adapter sequence was trimmed off from the adaptor reads by using Cutadapt with parameters: "--adapter = 'TGGAATTCTCGGGTGCCAAGG'"[55]. Then, the trimmed reads were aligned to the mitochondrial genome, which was derived from the whole human genome, by using the Burrows-Wheeler Aligner (BWA)[56]. The A-site of the mapped reads was identified by Plastid with the parameters offset: 14, min_length: 25, max_length: 35[57]. All the parameters were set in the mito_config.yml file. Finally, the codon count table of all the samples was generated. The codon counts for each codon from all the mitochondrial DNA (mtDNA) encoded genes within each sample were summed up and ordered by the codon frequency in the mtDNA. The codon occupation was plotted based on the normalized codon counts. Stalling for each gene near the 3' end of the transcripts was quantified by comparing the codon occupancy of the last 15 codons of each gene. To eliminate the difference in transcript levels between the samples, the codon counts in individual genes were multiplied by the size factor. The size factor is derived from the average of the total codon counts divided by the total codon counts of each gene. The histogram of each gene was plotted based on these normalized codon counts.

## Northern blotting

RNA was isolated from cells using TRIZOL reagent (Thermo Fisher) according to the manufacturer's instructions with the following minor changes: RNA solution was incubated with isopropanol for 30 min, and the pellet was washed with absolute ethanol. RNA was solubilized at 4 °C in water containing 40 units of RNase block (Agilent Technologies). Samples for northern blotting, containing 3.5–4 µg total cell RNA in 1:1–1:1.5 (v:v) mixture of water and NorthernMax-Gly Sample Loading Dye (ThermoFisher), were heated for 10 min at 55 °C and cooled down on ice for 2–10 min. RNA was subsequently loaded and separated on a gel, containing 1.2% (w/v) agarose (Invitrogen), 1× NorthernMax Running Buffer (Thermo Fisher), 0.4 M formaldehyde and 0.3–0.45% methanol (Merck), in 1× NorthernMax Running Buffer (Thermo Fisher) for 5–7 h at 2.5 V/cm and 4 °C. Loading was assessed by visualizing 18 S rRNA bands under UV-light since the loading dye contains ethidium bromide.

The resolved RNA was transferred to the water-activated Amersham Hybond-N + nylon membrane (Cytiva) by capillary forces overnight at room temperature. Transferred RNA was then crosslinked to the membrane by exposing to UV-light (UVC 500 Crosslinker, Amersham Biosciences, 254 nm, 120 mJ/cm²). The membrane was either

directly hybridized with a radioactively labeled probe or stored dry at room temperature before use. DNA templates were obtained by PCR on human mitochondrial genomic DNA using gene-specific primers (listed in Supplementary Table 1). For this purpose, mitochondria were isolated (section "isolation of mitochondria"), and DNA was purified using DNeasy Blood & Tissue Kit (Qiagen). After PCR completion, products were separated on an agarose gel and extracted using QIAquick Gel Extraction Kit (Qiagen).

For gene-specific probe preparations (except ND6-specific probe), Prime-It II random primer labeling kit (Agilent Technologies) was used. First, 20–30 ng of DNA template in 26.5 µl of water were mixed with 10 µl of random oligonucleotide primers (provided in the kit) and heated at 95 °C for 5 min followed by cooling down on ice for 2 min. Next, 10 µl of 5× dCTP buffer and 1 µl of 5 U/µl Exo(-) Klenow enzyme (both provided in the kit) were added together with 20–30 µCi [α-$^{32}$P]-dCTP (PerkinElmer), and the mixture was incubated at 37 °C for 30 min followed by heating at 95 °C for 5 min and cooling down on ice.

For ND6-specific probe synthesis, Riboprobe Combination System – SP6/T7 kit (Promega) was used according to the manual except that reaction volume was 50 µl, [α-$^{32}$P]-rUTP (PerkinElmer) was used for labeling, and no cold rUTP was added to the reaction. Probe synthesis was performed at 37 °C for 2.5 h followed by cooling down on ice.

The radioactively labeled probe was then purified using Amersham MicroSpin G-50 column (Cytiva) and stored at −20 °C. UV-crosslinked membrane was pre-activated in hybridization buffer (Sigma) at 65 °C for at least 10 min. The purified radioactively labeled probe (heated at 95 °C for 5 min and cooled down on ice for every probe except ND6-specific probe) was added to 25–30 ml of the hybridization buffer and incubated with the membrane at 65 °C for 5 h up to overnight. The membrane was washed at least three times using 1× SSC buffer (150 mM NaCl, 15 mM trisodium citrate) supplemented with 0.1% (w/v) SDS for at least 10 min at 65 °C, covered in a plastic film, and exposed to storage Phosphor screens (Fujifilm). The signals were then visualized with Typhoon FLA 7000 Phosphorimager (GE Healthcare). Band intensities were quantified using ImageJ software. If necessary, hybridized probe was removed from the membrane before the next round of hybridization by boiling stripping buffer (0.05× SSC buffer, 0.01 M Na-EDTA pH 8, 0.1% (w/v) SDS). Uncropped blots are depicted in source data file.

### SDS-PAGE and western blotting

Protein samples were resuspended with 1× NuPAGE LDS sample buffer (Thermo Fisher) supplemented with 100 mM dithiothreitol, heated for 10 min at 75 °C, and separated on NuPAGE 4–12% Bis-Tris mini gels (Thermo Fisher) using NuPAGE 1× MES (Thermo Fisher) running buffer. Next, proteins were blotted onto PVDF membrane using 1× wet western blot buffer (40 mM glycine, 50 mM Tris and 20% methanol) at 4 °C for 2 h at 300 mA. Membranes were blocked for 1 h at RT with 5% (w/v) milk in Phosphate Buffered Saline with 0.1% Tween 20 (PBS-T). The blocked membranes were incubated overnight at 4 °C or for 1 h at RT with primary antibodies (listed in Supplementary Table 2) in PBS-T at indicated dilutions. Following incubation, membranes were washed three times with PBS-T and incubated with HRP-conjugated secondary antibodies (listed in Supplementary Table 2) in PBS-T for 1 h at RT. Membranes were washed three times with PBS-T. Detection was performed using Amersham ECL western blotting detection reagent (GE Healthcare, Amersham). Band intensities were quantified by ImageJ. Uncropped blots are depicted in source data file.

### De novo mitochondrial translation assay

Six-well plates were treated with 0.01% (w/v) poly-L-Ornithine in PBS for 1 h at 37 °C, washed three times with PBS and treated with 2 µg/ml laminin in PBS ON at 4 °C. The next day, surfaces were washed three times with PBS and 2 × 10⁵ cells (HEK WT and *mtRF1* KOs) were seeded per well and cultured for 48 h. Afterwards, cells were incubated twice for 5 min at 37 °C in Cys-/Met-free medium (DMEM, high glucose, no glutamine, no methionine, no cysteine, supplemented with 10% dialyzed fetal bovine serum, 1× GlutaMax, and sodium pyruvate) followed by 20 min incubation at 37 °C in in Cys-/Met-free medium supplemented with 100 µg/ml anisomycin to inhibit cytosolic translation. Subsequently, 1 ml Cys-/Met-free medium supplemented with 200 µCi of EasyTag EXPRESS [$^{35}$S] protein labeling mix (methionine and cysteine) (Perkin Elmer) were added to each dish and incubated for 45 min at 37 °C, 5% CO$_2$. Following labeling, cells were washed three times with 5 ml PBS, harvested by trypsination and consecutive centrifugation (5000 × g, 5 min, 4 °C), and stored at −20 °C. Cells were lysed by resuspension in 30 µl PBS supplemented with Complete EDTA-free protease inhibitor cocktail and 50 U Pierce universal nuclease (Thermo Fisher) and application of one freeze-thaw cycle. Protein contents were determined by Pierce BCA assay (Thermo Fishe). 1× NuPAGE LDS sample buffer (Thermo Fisher) was added to 30 µg lysate respectively and separated on NuPAGE 4–12 % Bis-Tris mini gels (Thermo Fisher). Coomassie staining was performed using Imperial Protein Stain (Thermo Fisher) according to manufactures suggestions. The gel was fixed in fixing solution (20% methanol, 7% acetic acid, 3% glycerol) for 1 h at RT and vacuum-dried at 65 °C for 2 h. The resultant gel was exposed to storage Phosphor screens (Fujifilm) and visualized with Typhoon FLA 7000 Phosphorimager (GE Healthcare). Band intensities were quantified by ImageJ.

### Transient siRNA mediated knockdown experiments

2 × 10⁵ cells (WT and *mtRF1* KO1) were transfected with 40 pmol of C12orf65-targeted (Thermo Fisher, ID: s40690), C6orf203-targeted (Thermo Fisher, ID: s27763), or non-targeted control (Thermo Fisher, cat# 4390843) siRNA using Lipofectamine RNAiMAX Transfection Reagent (Thermo Fisher) according to manufacturer's suggestions. 72 h after transfection, de novo mitochondrial translation was tested by [$^{35}$S]-labeling. Knockdown efficiency was tested by western blotting.

### OXPHOS activity assay

The respiratory chain enzyme activities in isolated mitochondria were determined as previously described[58]. 500 µg isolated mitochondria were resuspended in 100 µl resuspension buffer (250 mM sucrose, 15 mM KH$_2$PO$_4$, 2 mM MgAc$_2$, 0.5 mM EDTA, 0.5 g/l HSA pH 7.2). 10 µl aliquots were prepared and frozen at −80 °C. All assays were performed in duplicate at 37 °C using an Indiko automated photometer (Thermo Fisher Scientific) fitted with filters for 340, 405, 550, and 600 nm (bandwidth ±5 nm).

NADH:coenzyme Q reductase (complex I) and NADH:cytochrome c reductase (complex I + III): For the pretreatment, 590 µl of a solution, consisting of 5 mM KH$_2$PO$_4$, 5 mM MgCl$_2$, 0.5 g/l HAS (pH 7.2), was added to 10 µl frozen mitochondrial suspension. Within 1 min this was mixed with another 50 µl of the same solution supplemented with 7.15 g/l saponin.

For determination of complex I activity, pretreated mitochondria (72 µl) were incubated for 7 min in a reaction mixture with the following final composition: 50 mM KH$_2$PO$_4$, 5 mM MgCl$_2$, 5 g/l HSA, 0.2 mM KCN, 1.2 mg/l antimycin A, and 0.12 mM coenzyme Q1 (pH 7.5). NADH was added to a final concentration of 0.15 mM, and the decrease in absorbance was monitored at 340 nm for 1 min before and after the addition of 2 mg/l rotenone. The final volume was 150 µL. The rotenone-sensitive activity was calculated with the use of an extinction coefficient of 6.81 l/mmol/cm. For determination of complex I + III activity, pretreated mitochondria (6 µL) were incubated for 7 min in a reaction mixture with the following final composition: 50 mM KH$_2$PO$_4$, 5 mM MgCl$_2$, 5 g/l HAS, 0.2 mM KCN, and 0.12 mM cytochrome c (oxidized form) (pH 7.5). NADH was added to a final concentration of 0.15 mM, and the increase in absorbance was monitored at 550 nm for 1 min before and after the addition of rotenone, 2 mg/L. The final volume was 125 µl. The rotenone-sensitive activity was calculated.

Succinate dehydrogenase (complex II) and Succinate:cytochrome c reductase (complex II + III): For the pretreatment, 10 µl frozen mitochondrial suspension was incubated at 37 °C for 30 min in 100 µl of a solution consisting of 50 mM $KH_2PO_4$, 30 mM succinate, 7.5 mM $MgCl_2$, and 0.45 g/l saponin (pH 7.2). Complex II activity was determined as previously described[59]. Pretreated mitochondria (10 µl) were incubated for 15 min in a reaction mixture with the following final composition: 20 mM $KH_2PO_4$, 5 mM $MgCl_2$, 25 mM succinate, 0.2 mM KCN, 0.05 mM 2,6-dichloroindophenol (DCIP), and 2 mg/l antimycin A (pH 7.5). The blank rate was measured for 1 min. Coenzyme $Q_1$ was added to a final concentration of 0.05 mM, and the decrease in absorbance was monitored at 600 nm for 1 min. The final volume was 150 µL. Activity was calculated with the use of an extinction coefficient of 22 l/mmol/cm

For determination of complex II + III activity, the blank rate was measured in the reagent, consisting of 50 mM $KH_2PO_4$, 5 mM $MgCl_2$, 5 g/l HSA, 0.2 mM KCN, 30 mM succinate, 2 mg/l rotenone, and 0.12 mM cytochrome c (oxidized form) (pH 7.5). Pretreated mitochondria (5 µl) were added, and the enzyme-catalyzed reduction of cytochrome c was monitored at 550 nm for 2 min. Final volume was 150 µl.

Cytochrome c oxidase (complex IV): For the pretreatment, mitochondria were diluted to a concentration corresponding to 100 U/l of citrate synthase in a solution containing 1 g/l digitonin, and 50 mM $KH_2PO_4$ (pH 7.5). 10 µl frozen mitochondrial suspension was diluted to a final volume of 70–250 µl, depending on the samples concentration of citrate synthase. The blank rate was recorded in the reagent, consisting of 50 mM $KH_2PO_4$, 2 mg/l rotenone, and 0.03 mM cytochrome c (reduced form) (pH 7.5). Pretreated mitochondria (10 µl) were added, and the enzyme-catalyzed oxidation of cytochrome c was followed at 550 nm for 1 min. The final volume was 250 µl. Reduced cytochrome c was prepared using ascorbate.

Citrate synthase: CS was used as mitochondrial marker and was determined as previously described[60]. For the pretreatment, 240 µl of a solution, consisting of 50 mM $KH_2PO_4$, 1 mM EDTA, 0.1% Triton X-100 (pH 7.5), was added to 10 µl frozen mitochondrial suspension.

Pretreated mitochondria (20 µl) were incubated for 5 min in a reaction mixture with the following final composition: 50 mM Tris, 0.20 mM 5,5′-Dithiobis(2 nitrobenzoic acid) (DTNB), 0.1 mM Acetyl-CoA (pH 8.1). Oxaloacetic acid was added to a final concentration of 0.5 mM, and the increase in absorbance was monitored at 405 nm for 1 min. The final volume was 250 µL. Activity was calculated with the use of an extinction coefficient of 13.6 l/mmol/cm.

## BN PAGE + complex activity

All steps were performed on ice or at 4 °C if not otherwise stated. 50 µg mitochondria were lysed in 25 µl lysis buffer (20 mM Tris pH7.4, 0.1 mM EDTA, 10% glycerol, 1.5 w/v digitonin, 1× Complete EDTA-free protease inhibitor cocktail) for 20 min. After lysis, insoluble material was removed by centrifugation (30 min, 17,000 × g). Next, 20 µl of supernatants were transferred into a new tube, supplemented with blue native loading dye (5% Coomassie blue G, 500 mM e-amino n-caproic acid in 100 mM Bis Tris pH 7.0) and loaded onto native PAGE 3–12% (Thermo Fisher). Gels were run at 150 V using NativePAGE Running buffer kit (Thermo Fisher). The activity of mitochondrial respiratory complexes was measured in-gel by addition of buffers containing OXPHOS substrates specific for Complex I (2 mM Tris pH 7.4, 0.1 mg/ml NADH, 2.5 mg/ml iodonitrotetrazolium chloride) or Complex IV (0.5 mg/ml diaminobenzidine (DAB), 50 mM phosphate buffer pH 7.4, 1 mg/ml cytochrome C, 0.2 M sucrose, 20 µg/ml catalase) at RT. Alternatively, gels were washed 15 min in 1× wet western blot buffer (40 mM glycine, 50 mM Tris and 20 % methanol) supplemented with 1% SDS and proteins were transferred onto PVDF membrane by western blotting. Before antibody incubation, membranes were destained by three 1 min washes with methanol.

## Isolation of mitoribosomes from bovine liver mitochondria

Bovine liver mitochondria were isolated according to a previously published protocol[61] and stored at −80 °C in 20 g aliquots until further use. For purification of mitoribosomes, 20 g mitochondria were resuspended in mitochondria isolation buffer (25 mM HEPES/KOH pH 7.5, 100 mM KCl, 20 mM Mg(OAc)₂, 70 mM sucrose, 210 mM mannitol, 2 mM DTT, 1× Complete EDTA-free protease inhibitor cocktail (Roche)) and treated with 10 U/ml DNase I for 20 min on ice. Next, mitochondria were pelleted by centrifugation at 10,000 × g for 10 min at 4 °C and resuspended in 200 ml mitochondria isolation buffer. A freshly prepared solution of 20 mg/ml digitonin in mitochondria isolation buffer was added to a final concentration of 0.1% digitonin. The sample was incubated on ice for 5 min and mitoplasts were pelleted by centrifugation at 10,000 × g for 10 min at 4 °C. The mitoplasts were then washed with 200 ml mitochondria isolation buffer and pelleted by centrifugation at 10,000 × g for 10 min at 4 °C. Subsequently, mitoplasts were lysed by addition of two volumes of lysis buffer (25 mM HEPES pH 7.5, 100 mM KCl, 20 mM Mg(OAc)₂, 2% Triton X-100, 2 mM DTT, 1× Complete EDTA-free protease inhibitor cocktail (Roche)) and incubated on ice for 15 min. The sample was then centrifuged at 30,000 × g for 60 min. The supernatant was overlayed on top of an equal volume of 1.1 M sucrose in lysis buffer containing 1% Triton X-100 and centrifuged at 150,000 × g for 16 h at 4 °C to pellet mitoribosomes. Crude mitoribosomes were resuspended in ribosome isolation buffer (25 mM HEPES pH 7.5, 100 mM KCl, 20 mM MgOAc, 2 mM DTT) on ice. The sample was clarified by centrifugation at 10,000 × g for 10 min and overlaid on top of a 10–30% sucrose gradient prepared in ribosome isolation buffer in SW32 tubes. Sucrose gradients were centrifuged at 90,000 × g for 16 h at 4 °C and fractionated by monitoring the absorbance at 260 nm. Fractions corresponding to 55S mitoribosomes were pelleted at 160,000 × g for 16 h at 4 °C. The pellet was then resuspended in ribosome isolation buffer, flash-frozen and stored at −80 °C until further use.

## Generation and purification of recombinant proteins

Mitochondrial translation initiation factors mtIF2 and mtIF3 were overexpress and purified as previously described[62]. E. coli EF-Tu/EF-Ts was prepared as previously described in[63].

Human mitochondrial mtEFG1 was purified as previously described in ref. [64].

For generation of mtRF1, a codon-optimized construct (Genscript) corresponding to the mature form of human mtRF1 (amino acids 40–446) was cloned into a pETM-11 vector. mtRF1 was expressed in Arctic Express (DE3) E. coli cells (Agilent) at 16 °C for 20 h in Magic Media (Thermo Fisher Scientific). The protein was purified over a His-Select Ni2+ resin (Sigma-Aldrich) and dialyzed against H-0.5 (25 mM Tris-HCl pH 7.4, 0.5 mM EDTA, 10% glycerol, 1 mM dithiothreitol and 500 mM NaCl) after the addition of TEV protease at a 1:50 protease:protein ratio. The TEV protease was removed by a second Ni²⁺ resin and the flow through was purified over a HiLoad 16/60 Superdex 200 pg gel filtration column (GE Healthcare) in buffer H-0.5 lacking glycerol. After purification, mtRF1 was concentrated using a 10,000 MWCO Vivaspin-10 device (Sartorius) to 10 mg/ml. 5% glycerol was added to the protein before it was snap frozen in liquid nitrogen and stored at −80 °C.

For purification of mtRF1a, a construct corresponding to the mature form of human mtRF1a (amino acids 27–380) was cloned into pQlinkH vector (#13667, Addgene) resulting in an N-terminal poly-histidine tag followed by a TEV cleavage site. The plasmid was transformed into E. coli strain Rosetta 2 (DE3) (Agilent) for protein expression. Cells were cultured in Lysogeny Broth media at 37 °C until optical density of 0.6 at 600 nm was reached. Protein expression was induced by the addition of 0.1 mM final concentration of isopropyl-D-thiogalactoside. Cells were further grown for 16 h at 25 °C and harvested by centrifugation. The protein was purified over a His-select Ni²⁺

resin (Sigma-Aldrich). Further clean-up was performed by using a HiLoad 16/600 Superdex 200 pg gel filtration column (GE Healthcare) in protein buffer (20 mM Tris pH 7.4, 200 mM NaCl, 1 mM DTT). The fractions corresponding to human mtRF1a were pooled together and concentrated up to 2 mg/ml using a 10,000 MWCO Vivaspin-10 spin columns (Sartorius). Glycerol was added to a final concentration of 5% before snap freezing in liquid nitrogen and storing at −80 °C until usage.

### Isolation of tRNAs
tRNAs were isolated from either bulk *E. coli* MRE600 or bulk yeast tRNAs by hybridization to immobilized complementary oligoDNAs. fMet-tRNA and Gln-tRNA were purified from bulk *E. coli* tRNA, Phe-tRNA and Arg-tRNA were purified from bulk yeast tRNAs. Charging with the cognate fMet ($^{35}$S), Phe ($^3$H), Arg ($^3$H), and Gln ($^3$H) amino acids respectively, was performed as previously described[65].

### Leaderless mRNAs
Leaderless mRNAs were purchased from Integrated DNA Technologies. The sequences of all mRNAs used in this study are provided in Supplementary Table 1.

### Cytoplasmic PURE-LITE in vitro translation system
The PURE-LITE system[66] relies on the ability of the Cricket Paralysis IRES sequence to bind to 80S ribosomes and initiate translation without the addition of translation initiation factors. Therefore, this system allows for a detailed mechanistic investigation of cytoplasmic translation elongation in a relatively simple in vitro system. We prepared all reagents and the translation assay as described[66,67].

### Mitochondrial in vitro translation assay
All steps were performed in translation buffer, containing 25 mM HEPES (pH 7.5), 10 mM Mg(OAc)$_2$, 50 mM KCl, 1 mM DTT, 2 mM spermidine, 0.05 mM spermine, 1× Complete EDTA-free protease inhibitor cocktail and RNase inhibitors. A 50 µl reaction containing 600 nM 55S ribosomes, 1.2 µM mtIF3, 1.2 µM mtIF2, 1.2 µM mtEFG1, 1.2 µM EF-Tu/EF-Ts, 1.2 µM fM-tRNA$^{fM}$, 1.2 µM other tRNAs (depending on mRNA sequence (Supplementary Table 1)), 3 µM mRNA, and 1 mM GTP was incubated for 30 min at 37 °C. The reaction was immediately overlaid on top of a 1.1 M sucrose cushion prepared in translation buffer and centrifuged at 4345 × *g* for 3 h at 4 °C using a TLA 120.2 rotor. The ribosomal pellet (= POST complex) was then washed three times with 1 ml cold translation buffer and resuspended on ice in 50 µl cold translation buffer. The concentration of 55S ribosomes was determined by UV–VIS spectroscopy and the concentration of radioactive amino acid incorporated into the peptide was determined by scintillation counting.

### Thin layer electrophoresis
Reaction mixtures containing 0.2 µM POST3-complex mixtures with the tripeptide fMet-Phe-Arg-tRNA$^{Arg}$ in the ribosomal P-site (80 µL) were incubated with 0.8 M KOH at 37 °C for 1 h. Subsequently, the pH was reduced by addition of glacial acetic acid to a final concentration of 1.74 M. The samples were then lyophilized, resuspended in 5 µL water and centrifuged at 10,000 × *g* for 5 min at 4 °C to remove aggregates. 1 µL sample was spotted on a TLC-cellulose plate and electrophoresed in pyridine acetate buffer (pH 2.8) at 500 V for 2 h. The [$^{35}$S]-fMet-Phe-Arg tripeptide was visualized by phosphorimaging.

### Mitochondrial translation termination assay
POST complexes containing fMQ*-tRNA$^Q$ in the P-site and stop codons in the A-site were prepared as described above, using mRNAs from Supplementary Table 1. Radioactive Q was used for detection of the dipeptide during the release experiment. The release reaction was prepared in translation buffer by incubating 100 nM POST complex with 200 nM release factors mtRF1 or mtRF1a and 1 mM GTP in a total volume of 50 µL for 5 min at 37 °C. 100 pmol 70 S ribosomes were then added as carrier. The sample was immediately overlaid on top of a 1.1 M sucrose cushion prepared in translation buffer and centrifuged at 4345 × *g* for 3 h at 4 °C using a TLA 120.2 rotor. The supernatant was collected for later measurements and the pellet was washed 3 times with 1 ml cold translation buffer and finally resuspended in 100 µl translation buffer on ice. The ($^3$H)-radioactivity of the supernatant (released peptide) and pellet (non-released peptide) was measured by scintillation counting. The release activity of the translation termination factors was determined by dividing the radioactive counts of the supernatant to the total counts measured in the supernatant and the pellet.

### Quantification and statistical analysis
Quantification analysis of Figs. 2b, d, f, h, 3c, 4b, 5a, c and Supplementary Fig. 3 are derived from the mean values. Error bars represent the standard deviation. Statistical analysis was done by a two-tailed, unpaired t-test using GraphPad Prism version 9.4.1. The significance threshold was set at $P < 0.05$; indicated as * for $P < 0.05$, ** for $P < 0.01$ and *** for $P < 0.001$. Exact $P$-values are listed in the source data file. Image analysis was performed in ImageJ version 1.52.

### Reporting summary
Further information on research design is available in the Nature Portfolio Reporting Summary linked to this article.

## Data availability
Ribosome profiling data are deposited at ArrayExpress under accession number: E-MTAB-11687. The mass spectrometry proteomics data have been deposited to the ProteomeXchange Consortium via the PRIDE[68] partner repository with the dataset identifier PXD034342. Source data are provided with this paper. Any additional information required to reanalyze the data reported in this paper is available from the lead contact upon request. Source data are provided with this paper.

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

## Acknowledgements
This work was supported by Knut and Alice Wallenberg Foundation (WAF 2017 and KAW 2018.0080), Karolinska Institute and Max Planck Institute. A.K. is supported by Deutsche Forschungsgesellschaft (DFG, German Research foundation – 467373608). M.D.N. is supported by EMBO fellowship (ALTF-2020-606). C.R. is supported by NIH GM127374. J.R. would like to thank STIAS for providing research fellowship.

## Author contributions
A.K. and D.I.S. performed knock-out cell lines analysis. C.R. with the help of H.S. and M.D.N., performed in vitro studies. A.K., J.R., and Y.L. performed ribosome profiling analysis. R.W. performed spectro-photometric studies. I.A. performed proteomic analysis. A.K., D.I.S., C.R., H.S., M.D.N., J.R., Y.L., R.W., I.A., and B.C. contributed to the data interpretation and final version of the manuscript.

## Funding

## Competing interests
The authors declare no competing interests.
