## [Peer Review File · Nature Communications]

Human mitochondria require mtRF1 for translation termination at non-canonical stop codonsREVIEWER COMMENTS

Reviewer #1 (Remarks to the Author):

The work of Kruger et al. aims to solve the lingering question in the field of mitochondrial gene expression, the role of mtRF1 release factor in the termination of non-canonically ended mitochondrial transcripts, COX1 and ND6. Indeed, despite over two decades passed from initial discovery, the exact functional contribution and specificity of mtRF1 have been a matter of longstanding debate. This very well-written and important manuscript by Kruger et al. provides several critical pieces of biochemical evidence that shed light on the termination of mitochondrial translation and the role of mtRF1 in this process. It also proves the critical mean of mtRF1 for releasing both alternatively terminated transcripts in human mitochondria. Moreover, to address their questions, the authors developed a beautiful and versatile in vitro system that, in the future, will allow the understanding of many yet uncharted molecular mechanisms underlying mitochondrial translation. The experimental design is well-tailored to answer major questions of this manuscript, and the authors deliver high-quality data that support their major conclusions. Indisputably, the work by Kruger et al. will be of interest to a broad community focused on mitochondrial gene expression and regulation of mitochondrial function. Although I greatly support the manuscript being published in Nature Communications, some minor points could be still considered by the authors:

1) Although in the first part of the manuscript authors carefully address the function of mtRF1 in the regulation of both COX1 and ND6 translation, in the later parts, they only focus on COX1. It is not clearly explained why the authors do not follow ND6 regulation further. Does it mean that although mtRF1 participates in the termination of ND6 translation, its function should be rather neglected? Could authors speculate more about the mtRF1/ND6 interplay? Is the overall mild effect of mtRF1 depletion on ND6 (transcript and protein) levels universal, or are there still particular contexts that may require more substantial reliance on the mtRF1/ND6 axis (e.g., tissue specificity, pathological circumstances)?

2) The authors discuss the plausible contribution of other release factors to the regulation of non-canonically terminated transcripts, particularly C12orf65 and ICT1 (with less likely involvement of ICT1). Do levels of these factors actually change upon mtRF1 deletion? Would knockdown of C12orf65 in the mtRF1 background worsen the synthesis of ND6 and/or COX1 subunits?

3) I fully agree with the authors that accelerated degradation of COX1 transcripts from stalled mitoribosomes could be responsible for strongly decreased COX1 mRNA levels. To further support this hypothesis with evidence, would it be possible to enhance the stability of COX1 transcripts by transient downregulation of SUV3 / PNPase complex? Would it improve the overall production of COX1 subunits? In line, is the LRPPRC/SLIRP machinery changed upon depletion of mtRF1?

Reviewer #2 (Remarks to the Author):

The function of mtRF1 in vertebrates, and there with the genetic code of mitochondria have long been debated. The manuscript of Kruger et al provides strong evidence that mtRF1 is involved in the release of two genes that lack a canonical stop codon (even after adding a poly-A tail). In principle this is a big step forward in the elucidation of the mitochondrial genetic code, which surprisingly is still unsolved, and I strongly support publication of the manuscript in Nature Communications. Even if I disagree on the often too “mechanistic” interpretation of the results.

Some issues:

Abstract:

“Together, our results uncover the mechanism of translation termination in mitochondria”

I do not agree that the authors have uncovered a mechanisms of translation termination. In absence of translation termination translation stalls at these positions as there is no cognate tRNA. mtRF1 appears involved in the release, but the mechanism is (still) uncovered. Also the sentence “Our study thus provides a clear indication that mtRF1 can recognize AGA/AGG triplets as stop codons”(discussion) is imho too strong. Stop codons as we know them are specifically recognized by the release factors. The authors do not demonstrate such recognition.

Rest of the manuscript

Line 155: “Consequently, it can be assumed that AGA and AGG represent actual stop codons which are recognized by mtRF1” As the authors have only shown indirect evidence for this and no actual “recognition” I do not agree with this sentence.

“Such a substitution is likely to be well tolerated in a functional sense, since the used E. coli tRNAs have canonical secondary structures quite similar to the corresponding tRNAs in mammalian mitochondria” there are a number of mitochondrial tRNAs that have lost part of their secondary structure relative to E.coli, like tRNA(Ser) that has lost its D-arm, so I was surprised to read this.

“and also have some posttranscriptional modifications, which are present in the mito-tRNAs” The if you want to make that argument then you also have to question whether most/all human mttRNAs have modifications that are shared with E.coli.

“Because the required amounts of mitochondrial tRNAs are not currently available, we also used E. coli tRNAs” it is unclear which ones were used. I take it not the ones that recognize AGA/AGG?

As the authors can, as far as I understand, add tRNAs at will, would it be possible to repeat the experiment from figure 5C with another terminal codon for which no tRNA was provided? This would go a long way to support the specificity argument. I realize they already have UAG, but using AAA that is an evolutionary much more frequent terminal triplet at this position might be very interesting.

The authors mention that “a few species (e.g., *Rattus norvegicus*, *Mus musculus*, *Platichthys flesus*, and *Danio rerio*) have MTRF1 but lack AGA/AGG codons in their mitochondrial genome” This may have been the case when ref 18 was published, but by now, examining MT-CO1 in the ucsc genome browser show that having AGA/AGG at that position is actually quite rare among the vertebrates, and there is very little selection on maintaining any of the three nucleotides (except for having a purine at the second position). mtRF1 by contrast, appears to be universal. This casts doubt on any specific role of AGA/AGG in the release mechanism, specifically because no mechanism has been provided. (Do note that the authors have used the evolution arguments themselves in arguing against the universality of the frameshift mechanism)

Editorial:

Line 370 “the the”

The explanation of figure 1D in the legend is a bit mindboggling. I also do not understand whether and why the profiles were clustered (as they appear to be)

Reviewer #3 (Remarks to the Author):

This manuscript by the Rorbach group shows that the mitochondrial release factor mtRF1 is required for translation termination at non-standard stop codons AGA and AGG. Given its fundamental and biomedical implications, the discovery is highly relevant in the field of mitochondrial protein synthesis and in mRNA translation in general. The work relies on the analysis of an mtRF1 ko cell line using mitoribosome profiling, and a novel in vitro mitochondrial translation assay. The in vitro translation system uses bacterial translation factors and tRNAs. The system is working to prove the points regarding the role of mtRF1 as a release factor. I would suggest, however, to be a bit less vague when describing the use of thd3ese factors in sentences 351-355. In the KO mitoribosomes, COX1 and ND6 are inefficiently released, leading to a CIV assembly and function defect but not to a CI defect. The data obtained in the KO cell lines was recapitulated by reconstituting the KO with a non-functional variant of mtRF1 in which the GGQ motif was mutated. The manuscript is technically and conceptually sound and exciting.

Reviewer #4 (Remarks to the Author):

General Comments

In the manuscript NCOMMS-22-24287 “Human Mitochondria Require mtRF1 for translation termination at non-canonical stop codons”, the authors investigate the mechanism of translation termination in mitochondria. Overall I found this to be a well written manuscript. The focus of my review is focused on the mass spectrometry aspect of the manuscript. Overall the data is compelling, but the presentation is confusing. Clarification of figure labels and corresponding text will enhance this section.

Specific Comments

1. Figure 3A and the corresponding discussion on page 13 should be clarified. The X axis labels show KO1 and KO2 and are color coded to distinguish what complex it refers too. The text in the discussion describes the comparison between WT and mtRF1 KO cells. Please clarify in the text it is WT vs mtRF1 KO1 and WT vs mtRF KO2. Do “Complex 1” = NDUF8 and “Complex 2” = SDHB etc.? If so, it would be easier to read the figure if you labeled them as such. Complex suggests a complex of several proteins. Later in the method section there is a whole page (page 32) of perhaps the definition of complexes, but it does not make any sense at all.
2. In the Methods Section, please elaborate on which TMTpro kit was used and how, was WT light and KO heavy? Did KO1 and KO2 receive the same label? There are several variations of these labels.
3. On page 31, please include the Orbitrap Tribrid parameters used during the data analysis (resolution, target IG what was the fragmentation and fragmentation voltages etc).
4. Please elaborate on the settings used in the MaxQuant analysis.
5. What does Line 645 – 663mean? I can’t follow this section at all.

Response to Referees letter

We thank the Reviewers for taking the time to provide valuable suggestions on how to improve the study, data interpretation and presentation. We are happy to receive overall positive feedback from all four Reviewers.

Aside from addressing the Reviewers' requests, we introduced additional minor changes to improve the quality of our data and provided the missing information.

Changes unrelated to reviewer comments:

- Protein steady-state levels of ND6 were revised. Due to the low expression levels, high hydrophobicity, and small size of ND6, it is very challenging to detect ND6 by western blotting. During the revision process, we received a new antibody from Thermo Fisher, which gave a better signal and showed higher reproducibility. Therefore, we repeated the detection of ND6 steady-state levels in HEK WT and mtRF1 KO cells and included the new data in Figures 2G and H. The specificity of the antibody was tested by chloramphenicol treatment, which is now included in Suppl. Figure S3 B and described in the text (lines 205-214). By updating this data, we improved the quality of our data, but the results and conclusions were not affected.
Furthermore, we carefully investigated the position of ND6 in [³⁵S]-labeling using the new antibody (Suppl. Figure 3B). Figure 2E and 4C were updated accordingly. Results and conclusions are not affected by these changes.
- Addition of methods parts:
 - mtRF1a purification (lines 1026-1038)
 - thin layer chromatography (lines 1067-1074)
- Author list: Minh Duc Nguyen was added to the author list. He performed the purification of mtRF1a for the *in vitro* translation assay.

The reviewers' comments are addressed point-by-point in the following section.

Reviewer #1 (Remarks to the Author):

The work of Kruger et al. aims to solve the lingering question in the field of mitochondrial gene expression, the role of mtRF1 release factor in the termination of non-canonically ended mitochondrial transcripts, COX1 and ND6. Indeed, despite over two decades passed since initial discovery, the exact functional contribution and specificity of mtRF1 have been a matter of longstanding debate. This very well-written and important manuscript by Kruger et al. provides several critical pieces of biochemical evidence that shed light on the termination of mitochondrial translation and the role of mtRF1 in this process. It also proves the critical mean of mtRF1 for releasing both alternatively terminated transcripts in human mitochondria. Moreover, to address their questions, the authors developed a beautiful and versatile *in vitro* system that, in the future, will allow the understanding of many yet uncharted molecular mechanisms underlying mitochondrial translation. The experimental design is well-tailored to answer the major questions of this manuscript, and the authors deliver high-quality data that support their major conclusions. Indisputably, the work by Kruger et al. will be of interest to a broad community focused on mitochondrial gene expression and regulation of mitochondrial function. Although I greatly support the manuscript being published in Nature Communications, some minor points could be still considered by the authors:

1) Although in the first part of the manuscript authors carefully address the function of mtRF1 in the regulation of both COX1 and ND6 translation, in the later parts, they only focus on COX1. It is not clearly explained why the authors do not follow ND6 regulation further. Does it mean that although mtRF1 participates in the termination of ND6 translation, its function should be rather neglected? Could authors speculate more about the mtRF1/ND6 interplay? Is the overall mild effect of mtRF1 depletion on ND6 (transcript and protein) levels universal, or are there still particular contexts that may require more substantial reliance on the mtRF1/ND6 axis (e.g., tissue specificity, pathological circumstances)?

This is a very interesting and important point. Unfortunately, we do not have an answer to it yet. We analyzed the effect of mtRF1 depletion on ND6 and COX1 mRNA and protein levels and on complex I and IV activities. All data suggest that the absence of mtRF1-mediated release of ND6 is tolerated quite well. Yet, it is important to note that our study is performed in HEK cells, and we do not have any *in vivo* data yet. Furthermore, it is an interesting thought that this might change during stress or pathogenic conditions. We are currently studying the effect of mtRF1a depletion in HEK cells. Interestingly, we again see very different effects on different mitochondrial transcripts and proteins, supporting the hypothesis that mitochondria have transcript-specific downstream mechanisms. Potential reasons are mentioned in the discussion section and include: structures of the transcripts, differences in the incorporation of newly produced peptides into OXPHOS complexes, and, as this reviewer suggested, tissue-specific compensatory mechanisms (added in lines 541-546).

To put more emphasis on the effects of mtRF1 depletion on ND6 and complex I, we added some additional text in the result (lines 283-287; lines 300-304; lines 335-336) and discussion sections (lines 541-546).

2) The authors discuss the plausible contribution of other release factors to the regulation of non-canonically terminated transcripts, particularly C12orf65 and ICT1 (with less likely involvement of ICT1). Do levels of these factors actually change upon mtRF1 deletion? Would knockdown of C12orf65 in the mtRF1 background worsen the synthesis of ND6 and/or COX1 subunits?

This is a very relevant question and we tried to address it through several experiments (protein steady-state levels by western blotting and siRNA-mediated knockdown experiments of C12orf65 and C6orf203). Interestingly, we neither see significant upregulation of any of these factors nor a specific effect on translation in mtRF1 KO cells upon downregulation (new data is included in Suppl. Figure S4, main text (lines 216-241), and discussion (lines 495-498)). As these experiments represent, however, only preliminary analysis in this direction, we still cannot rule out their involvement in rescuing stalled mitoribosomes in mtRF1 KO cells. It is e.g., not clear why knockdown of C6orf203/C12orf65 has only minor effects on *de novo* translation, while complete loss of functional C6orf203/C12orf65 results in a significant (~50 %) decrease of all *de novo* translated proteins [1,2]. Therefore, this research topic requires further investigations, which however goes beyond the scope of this study.

1. Antonicka, H. *et al.* Mutations in C12orf65 in Patients with Encephalomyopathy and a Mitochondrial Translation Defect. *Am. J. Hum. Genet.* **87**, 115–122 (2010).
2. Gopalakrishna, S. *et al.* C6orf203 is an RNA-binding protein involved in mitochondrial protein synthesis. *Nucleic Acids Res.* **47**, 9386–9399 (2019).

3) I fully agree with the authors that accelerated degradation of COX1 transcripts from stalled mitoribosomes could be responsible for strongly decreased COX1 mRNA levels. To further support this hypothesis with evidence, would it be possible to enhance the stability of COX1 transcripts by transient downregulation of SUV3 / PNPase complex? Would it improve the overall production of COX1 subunits? In line, is the LRPPRC/SLIRP machinery changed upon depletion of mtRF1?

The authors agree with this reviewer that SUV3/PNPase are good candidates for the degradation of COX1 mRNA. Yet, studying their role in COX1 mRNA degradation is challenging as SUV3/PNPase are not only involved in RNA degradation but also RNA processing. Loss of functional SUV3/PNPase was associated with changes in mitochondrial RNAs [1,2,3] and loss of mitochondrial translation [4]. Therefore, the authors believe that knockdown experiments would not provide a clear answer on the involvement of SUV3/PNPase in COX1 degradation.

1. Borowski, Lukasz S., Andrzej Dziembowski, Monika S. Hejnowicz, Piotr P. Stepień, and Roman J. Szczesny. 2013. "Human Mitochondrial RNA Decay Mediated by PNPase–HSuv3 Complex Takes Place in Distinct Foci." *Nucleic Acids Research* **41** (2): 1223. <https://doi.org/10.1093/NAR/GKS1130>.
2. Szczesny, Roman J., Lukasz S. Borowski, Lien K. Brzezniak, Aleksandra Dmochowska, Kamil Gewartowski, Ewa Bartnik, and Piotr P. Stepień. 2010. "Human Mitochondrial RNA Turnover Caught in Flagranti: Involvement of HSuv3p Helicase in RNA Surveillance." *Nucleic Acids Research* **38** (1): 279–98. <https://doi.org/10.1093/NAR/GKP903>.
3. Slomovic, Shimyn, and Gadi Schuster. 2008. "Stable PNPase RNAi Silencing: Its Effect on the Processing and Adenylation of Human Mitochondrial RNA." *RNA* **14** (2): 310–23. <https://doi.org/10.1261/RNA.697308>.

4. Khidr, Lily, Guikai Wu, Antonio Davila, Vincent Procaccio, Douglas Wallace, and Wen Hwa Lee. 2008. "Role of SUV3 Helicase in Maintaining Mitochondrial Homeostasis in Human Cells." *The Journal of Biological Chemistry* 283 (40): 27064. <https://doi.org/10.1074/JBC.M802991200>.

As suggested by the reviewer, we tested LRPPRC and SLIRP levels in mtRF1 KO cells compared to WT cells. Both proteins were present in our mass spectrometry dataset from isolated mitochondria. While SLIRP levels were not significantly affected in both mtRF1 KO clones, one of the clones (mtRF1 KO2) showed a slight downregulation in LRPPRC. This was further confirmed by western blotting (Figure i; below). LRPPRC levels have previously been shown to correlate with mitochondrial RNA levels [5]. Consequently, decreased levels of LRPPRC might reflect decreased levels of COX1 transcript. The mild difference between the two clones might be caused by the different compensatory mechanisms of both clones.

5. Ruzzenente, Benedetta, Metodi D. Metodiev, Anna Wredenberg, Ana Bratic, Chan Bae Park, Yolanda Cámara, Dusanka Milenkovic, et al. 2012. "LRPPRC Is Necessary for Polyadenylation and Coordination of Translation of Mitochondrial MRNAs." *EMBO Journal* 31 (2): 443–56. <https://doi.org/10.1038/emboj.2011.392>.

Table i: Quantitative mass spectrometry analysis of mitochondrial lysates from mtRF1 KO1 and KO2 compared to WT cells.

Gene name	logFC	Adjusted P-value	Significant	Clone
SLIRP	0.249	0.203	NA	KO1
LRPPRC	-0.215	0.373	NA	KO1
SLIRP	-0.274	0.116	NA	KO2
LRPPRC	-0.572	1.480	+	KO2

Figure i: LRPPRC steady-state levels. Left: LRPPRC protein levels analyzed by western blotting. Mitochondrial lysates were loaded. A representative blot is depicted. Loading was assessed by SDHA detection. Right: Quantification normalized to protein levels in WT cells. Means and SD of n=3 biological replicates. Unpaired t-test (* P<0.05; ** P<0.01; *** P<0.001).

Reviewer #2 (Remarks to the Author):

The function of mtRF1 in vertebrates, and there with the genetic code of mitochondria have long been debated. The manuscript of Kruger et al provides strong evidence that mtRF1 is involved in the release of two genes that lack a canonical stop codon (even after adding a poly-A tail). In principle this is a big step forward in the elucidation of the mitochondrial genetic code, which surprisingly is still unsolved, and I strongly support publication of the manuscript in Nature Communications. Even if I disagree on the often too “mechanistic” interpretation of the results.

We would like to thank the reviewer for the overall positive comments and suggestions.

Abstract:

“Together, our results uncover the mechanism of translation termination in mitochondria”
I do not agree that the authors have uncovered a mechanisms of translation termination. In absence of translation termination translation stalls at these positions as there is no cognate tRNA. mtRF1 appears involved in the release, but the mechanism is (still) uncovered. Also the sentence “Our study thus provides a clear indication that mtRF1 can recognize AGA/AGG triplets as stop codons”(discussion) is imho too strong. Stop codons as we know them are specifically recognized by the release factors. The authors do not demonstrate such recognition.

Abstract:

We agree that the mechanism of translation termination was not completely uncovered as there are still open questions. The sentence (line 31) was changed accordingly.

Discussion:

We are aware that the direct recognition was not demonstrated (this could only be done by structural studies), but we believe that our data is still strong enough to draw this conclusion. We used two very different and independent methodological approaches, where we showed that mtRF1 is important in releasing at AGA/AGG codons. In our *in vitro* translation assay we observe the release of a dipeptide in the presence of mtRF1 only when AGA or AGG are placed in the A-site, but not when UAA or UAG are present. Considering that mtRF1 contains all domains, which are needed to be a canonical release factor (GGQ domain and codon recognition domain), the most likely scenario is that mtRF1 acts as a release factor. The fact that mtRF1a is a canonical release factor recognizing UAA/UAG has also been first demonstrated by *in vitro* translation assays [1,2] and could only recently be confirmed by structural studies [3].

1. Nozaki, Yusuke, Noriko Matsunaga, Toshihiro Ishizawa, Takuya Ueda, and Nono Takeuchi. 2008. “HMRF1L Is a Human Mitochondrial Translation Release Factor Involved in the Decoding of the Termination Codons UAA and UAG.” *Genes to Cells* 13 (5): 429–38. <https://doi.org/10.1111/j.1365-2443.2008.01181.x>.
2. Soleimanpour-Lichaei, Hamid Reza, Inge Kühn, Mauricette Gaisne, Joao F. Passos, Mateusz Wydro, Joanna Rorbach, Richard Temperley, et al. 2007. “MtRF1a Is a Human Mitochondrial Translation Release Factor Decoding the Major Termination Codons UAA and UAG.” *Molecular Cell* 27 (5): 745–57. <https://doi.org/10.1016/j.molcel.2007.06.031>.

3. Kummer, Eva, Katharina Noel Schubert, Tanja Schoenhut, Alain Scaiola, and Nenad Ban. 2021. "Structural Basis of Translation Termination, Rescue, and Recycling in Mammalian Mitochondria." *Molecular Cell* 81 (12): 2566-2582.e6. <https://doi.org/10.1016/j.molcel.2021.03.042>.

Rest of the manuscript

Line 155: "Consequently, it can be assumed that AGA and AGG represent actual stop codons which are recognized by mtRF1" As the authors have only shown indirect evidence for this and no actual "recognition" I do not agree with this sentence.

We agree that at this stage of the study the conclusion is too strong. We, therefore, deleted this sentence.

"Such a substitution is likely to be well tolerated in a functional sense, since the used *E. coli* tRNAs have canonical secondary structures quite similar to the corresponding tRNAs in mammalian mitochondria" there are a number of mitochondrial tRNAs that have lost part of their secondary structure relative to *E. coli*, like tRNA(Ser) that has lost its D-arm, so I was surprised to read this.

"and also have some posttranscriptional modifications, which are present in the mito-tRNAs" The if you want to make that argument then you also have to question whether most/all human mttRNAs have modifications that are shared with *E. coli*.

This is a valuable point and indeed, the structures of several mitochondrial tRNAs are strikingly different from their bacterial counterparts. However, we used a limited set of tRNAs in our study: *E. coli* tRNA^{fMet}, yeast tRNA^{Phe} and yeast tRNA^{Arg} for the characterization of the mitochondrial *in vitro* translation system, and *E. coli* tRNA^{fMet} and tRNA^{Gln} for the translation termination experiment. These tRNAs possess secondary structures similar to their human mitochondrial counterparts. To demonstrate the similarity, we included here the aligned structures of *E. coli* tRNA^{fMet} (PDB 6O7K) and human mitochondrial tRNA^{Met} from translation initiation complexes (PDB 7PO2) (Figure ii). Furthermore, we compared in detail the tRNA modifications present in both tRNAs (Figure iii). Even though there are substantial differences between the bacterial and mitochondrial tRNA modifications, most of them have only stabilizing purpose (such as Ψ , D, T, s4U, Cm, m7G) and are therefore unlikely to affect binding to the mitoribosome. Yet, modifications in the anti-codon loop are likely to affect codon binding specificity and should therefore be looked at more carefully. While mitochondrial tRNA^{Met} possesses f5C34, this modification is lacking in bacteria. The modification allows recognition of mitochondria-specific AUA start codon. As we were using only the canonical start codon AUG in our *in vitro* experiments, this modification is not essential. The anti-codon loop of tRNA^{Gln} is modified in bacteria (cmnm5s2U34) as well as in mitochondria (τ m5s2U34). As both modifications allow correct base pairing with 3'-A-ending codons, such as CAA used in this study, there is likely no difference between using bacterial or mitochondrial tRNA. In conclusion, we can assume that not all modifications are essential for functionality, but it strongly depends on the exact location. Therefore, we think that the sentence "and also have some posttranscriptional modifications, which are present in the mito-tRNAs" is not relevant and changed this section accordingly (383-387).

Figure ii: Comparison of *E. coli* tRNA^{fMet} (PDB 6O7K) and human mitochondrial tRNA^{Met} (PDB 7PO2) tertiary structures from translation initiation complexes.

Figure iii. Comparison of modification profiles for human mitochondrial and *E. coli* tRNA^{Met} / tRNA^{fMet} (A, B) and tRNA^{Gln} (C, D). Shown modifications are listed in [1, 2]. A and C are modified from [1], B is modified from [3], D is modified from [4].

1. Suzuki, Takeo, Yuka Yashiro, Ittoku Kikuchi, Yuma Ishigami, Hironori Saito, Ikuya Matsuzawa, Shunpei Okada, et al. 2020. "Complete Chemical Structures of Human Mitochondrial tRNAs." *Nature Communications* 11 (1). <https://doi.org/10.1038/s41467-020-18068-6>.
2. Boccaletto, Pietro, Filip Stefaniak, Angana Ray, Andrea Cappannini, Sunandan Mukherjee, Elżbieta Purta, Małgorzata Kurkowska, et al. 2022. "MODOMICS: A Database of RNA Modification Pathways. 2021 Update." *Nucleic Acids Research* 50 (D1): D231–35. <https://doi.org/10.1093/NAR/GKAB1083>.
3. Stortchevoi, Alexei, Umesh Varshney, and Uttam L. RajBhandary. 2003. "Common Location of Determinants in Initiator Transfer RNAs for Initiator-Elongator Discrimination in Bacteria and in Eukaryotes." *Journal of Biological Chemistry* 278 (20): 17672–79. <https://doi.org/10.1074/JBC.M212890200>.
4. Liu, Ru Juan, Tao Long, Mi Zhou, Xiao Long Zhou, and En Duo Wang. 2015. "tRNA Recognition by a Bacterial tRNA^{Xm32} Modification Enzyme from the SPOUT Methyltransferase Superfamily." *Nucleic Acids Research* 43 (15): 7489–7503. <https://doi.org/10.1093/NAR/GKV745>.

"Because the required amounts of mitochondrial tRNAs are not currently available, we also used *E. coli* tRNAs" it is unclear which ones were used. I take it not the ones that recognize AGA/AGG?

Yes, indeed, we have chosen tRNA^{fMet} and tRNA^{Gln} that do not recognize AGA/AGG. The rationale is explained above. To make clearer, which tRNAs were used, we added information in the results (lines 383-385) and methods section (lines 1039-1044)

As the authors can, as far as I understand, add tRNAs at will, would it be possible to repeat the experiment from figure 5C with another terminal codon for which no tRNA was provided? This would go a long way to support the specificity argument. I realize they already have UAG, but using AAA that is an evolutionary much more frequent terminal triplet at this position might be very interesting.

As discussed with the reviewer during the revision process, this experiment is not essential for the manuscript's conclusion and therefore has not been performed.

The authors mention that "a few species (e.g., *Rattus norvegicus*, *Mus musculus*, *Platichthys flesus*, and *Danio rerio*) have MTRF1 but lack AGA/AGG codons in their mitochondrial genome" This may have been the case when ref 18 was published, but by now, examining MT-CO1 in the ucsc genome browser show that having AGA/AGG at that position is actually quite rare among the vertebrates, and there is very little selection on maintaining any of the three nucleotides (except for having a purine at the second position). mtRF1 by contrast, appears to be universal. This casts doubt on any specific role of AGA/AGG in the release mechanism, specifically because no mechanism has been provided. (Do note that the authors have used the evolution arguments themselves in arguing against the universality of the frameshift mechanism)

We thank this reviewer for carefully checking the presence/absence of AGA/AGG codons in mt-CO1 in a current database and noticing a discrepancy compared to data from 2012. To look deeper into this, we now have performed the analysis of 3'-terminal positions of mtDNA-encoded genes in vertebrates (6564 species with GenBank records for mitochondrial genome) using the Organelle Genome Resources portal provided by NCBI (<https://www.ncbi.nlm.nih.gov/genome/organelle/>). Across all analyzed mitochondrial

genomes, we found that 56 % carried at least one AGA/G stop codon at the 3'-end of an mtDNA-encoded gene (Figure iv, not only in COX1). Within this 56%, more than half could not terminate by -1 frameshift but would end up having a sense codon in the A-site. Interestingly, we found that even in closely related species there can be large variability in mtDNA stop codons – e.g., within the Haplorhini suborder of primates COX1 and ND6 3'-terminal positions are distributed across AGG, TAA, TAG, AGA. Therefore, we are convinced that there is an *in vivo* significance for AGA and AGG stop codons, which require another factor than mtRF1a for termination. We agree that it is surprising that not all organisms having mtRF1 also have AGA/AGG stop codons. Potential reasons are mentioned in our discussion section (lines 458-465).

Figure iv: Left: The presence of AGA/AGG stop codons in mitochondrial genomes from vertebrates with GenBank records for mitochondrial genome. Right: Codon distribution after -1 frameshifting (-1FS) in genomes having AGA/G stop codons. Analysis was carried out using data from the NCBI Organelle Genome Resources portal (<https://www.ncbi.nlm.nih.gov/genome/organelle/>). Sorting within 13620 GenBank records resulted in 6584 vertebrate records. 20 were discarded as they exhibited less than 13 genes (potential misannotation). Genomes were classified as “no AGA/G as stop” if all corresponding mt-ORFs were terminated with non-AGA/G codons.

Reviewer #3 (Remarks to the Author):

This manuscript by the Rorbach group shows that the mitochondrial release factor mtRF1 is required for translation termination at non-standard stop codons AGA and AGG. Given its fundamental and biomedical implications, the discovery is highly relevant in the field of mitochondrial protein synthesis and in mRNA translation in general. The work relies on the analysis of an mtRF1 ko cell line using mitoribosome profiling, and a novel in vitro mitochondrial translation assay. The in vitro translation system uses bacterial translation factors and tRNAs. The system is working to proof the points regarding the role of mtRF1 as a release factor. I would suggest, however, to be a bit less vague when describing the use of these factors in sentences 351-355. In the KO mitoribosomes, COX1 and ND6 are inefficiently released, leading to a CIV assembly and function defect but not to a CI defect. The data obtained in the KO cell lines was recapitulated by reconstituting the KO with a non-functional variant of mtRF1 in which the GGQ motif was mutated. The manuscript is technically and conceptually sound and exciting.

We would like to thank the reviewer for the positive evaluation of our manuscript.

The lack of clarity in sentences 351-355 was also pointed out by reviewer 2. We have now modified this text (new line numbering: 383-387). Please find the full explanation in the response to reviewer 2.

Reviewer #4 (Remarks to the Author):

General Comments

In the manuscript NCOMMS-22-24287 “Human Mitochondria Require mtRF1 for translation termination at non-canonical stop codons”, the authors investigate the mechanism of translation termination in mitochondria. Overall I found this to be a well written manuscript. The focus of my review is focused on the mass spectrometry aspect of the manuscript. Overall the data is compelling, but the presentation is confusing. Clarification of figure labels and corresponding text will enhance this section.

Specific Comments

1. Figure 3A and the corresponding discussion on page 13 should be clarified. The X axis labels show KO1 and KO2 and are color coded to distinguish what complex it refers too. The text in the discussion describes the comparison between WT and mtRF1 KO cells. Please clarify in the text it is WT vs mtRF1 KO1 and WT vs mtRF KO2. Do “Complex 1” = NDUF8 and “Complex 2” = SDHB etc.? If so, it would be easier to read the figure if you labeled them as such. Complex suggests a complex of several proteins. Later in the method section there is a whole page (page 32) of perhaps the definition of complexes, but it does not make any sense at all.

We thank the reviewer for pointing out this unclarity. We added tables in the supplementary material, listing all proteins included in the analysis (Suppl. Table 3-5). Furthermore, we changed the Figure legends: Figure 3A, Suppl. Figures S1C, and Suppl. Figure S4 and the Method section for clarification (lines 697-703). We did not label each individual protein in the box plots as it would have become too crowded.

2. In the Methods Section, please elaborate on which TMTpro kit was used and how, was WT light and KO heavy? Did KO1 and KO2 receive the same label? There are several variations of these labels.

We added the information on the TMTpro kit (lot number: WC314415) in the Method part (lines 649-652) as well as a table on the labeling scheme (Suppl. Table 6).

3. On page 31, please include the Orbitrap Tribrid parameters used during the data analysis (resolution, target IG what was the fragmentation and fragmentation voltages etc).

We added more details on the Orbitrap Tribrid parameter in the Method section (lines 670-682).

4. Please elaborate on the settings used in the MaxQuant analysis.

We added more information regarding the MaxQuant analysis (lines 685-695)

5. What does Line 645 – 663 mean? I can't follow this section at all.

We are sorry for the confusion. Instead of presenting the R script, we now added Suppl. Table 3-5 listing all proteins presented in the box plots. We changed the text in the Methods part accordingly (lines 697-703).

REVIEWERS' COMMENTS

Reviewer #1 (Remarks to the Author):

Many thanks to the authors for comprehensively addressing all my points (and supporting their conclusions with diligent experimental evidence). Additional figures and comments included in the revised manuscript clarify more on the role of mtRF1 in ND6 translation and complex I biogenesis next to the strong effect on the COX1/CIV. Meanwhile, careful discussion and further evaluations shed some light on the intriguing function of additional factors in regulating the fate of noncanonically terminated transcripts. Altogether, I am convinced that the findings of this work will be of great value to the field of mitochondrial gene expression and will open some new exciting directions for investigations. Therefore, I warmly support publishing this beautiful, important, high-quality story in Nature Communications.

Reviewer #2 (Remarks to the Author):

I am satisfied with the authors' reactions and modifications